# Controllable Unlearning for Image-to-Image Generative Models via $\varepsilon$-Constrained Optimization

**Xiaohua Feng**[1*]**, Yuyuan Li**[2*]**, Chaochao Chen**[1†]**, Li Zhang**[1]**, Longfei Li**[3]**,
Jun Zhou**[3]**, Xiaolin Zheng**[1]
[1]Zhejiang University, [2]Hangzhou Dianzi University, [3]Ant Group

## Abstract

While generative models have made significant advancements in recent years, they also raise concerns such as privacy breaches and biases. Machine unlearning has emerged as a viable solution, aiming to remove specific training data, e.g., containing private information and bias, from models. In this paper, we study the machine unlearning problem in Image-to-Image (I2I) generative models. Previous studies mainly treat it as a single objective optimization problem, offering a solitary solution, thereby neglecting the varied user expectations towards the trade-off between complete unlearning and model utility. To address this issue, we propose a controllable unlearning framework that uses a control coefficient $\varepsilon$ to control the trade-off. We reformulate the I2I generative model unlearning problem into a $\varepsilon$-constrained optimization problem and solve it with a gradient-based method to find optimal solutions for unlearning boundaries. These boundaries define the valid range for the control coefficient. Within this range, every yielded solution is theoretically guaranteed with Pareto optimality. We also analyze the convergence rate of our framework under various control functions. Extensive experiments on two benchmark datasets across three mainstream I2I models demonstrate the effectiveness of our controllable unlearning framework.

## 1 Introduction

Generative models have recently made significant progress in fields such as image recognition (Ho et al., 2020; Dhariwal & Nichol, 2021) and natural language processing (OpenAI, 2023; Touvron et al., 2023), capturing significant academic interest due to their boundless generative potential. Typically trained on vast datasets from the Internet, generative models inevitably assimilate latent biases and expose private information (Schwarz et al., 2021; Yang et al., 2025). Existing studies (Kuppa et al., 2021; Tirumala et al., 2022; Carlini et al., 2023; Xu et al., 2024) have revealed that generative models have a strong tendency to recall specific instances encountered during training, raising concerns that the models might output biases and leak private information when put into practical situations. Machine unlearning (Nguyen et al., 2022; Feng et al., 2024) presents a viable solution to address this issue. It aims to eliminate the knowledge learned from specific training data (forget set) while preserving the knowledge learned from the remaining data (retain set).

Implementing unlearning for generative models serves dual objectives, i.e., fulfilling privacy requirements and enhancing model reliability. On the one hand, legislation such as the General Data Protection Regulation (Voigt & Von dem Bussche, 2017) grants individuals the *right to be forgotten*. Consequently, service providers must unlearn specific private information from the model in response to an individual's request. On the other hand, the data available on the Internet is rife with biases and inaccuracies, which compromises model performance when used for training. By proactively unlearning the biased and inaccurate data, the service providers can improve the liability of their models.

---

[*]Equal contributions.
[†]Corresponding author (zjuccc@zju.edu.cn).

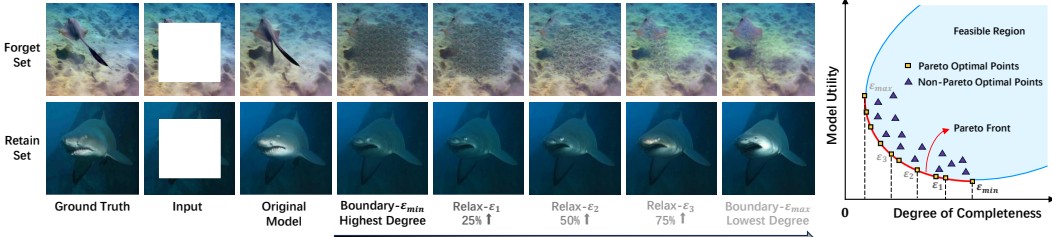

Figure 1: An overview of controllable unlearning. On the left, the first and second rows represent the forget set and the retain set, respectively. We first present the effect of unlearning in I2I generative models, followed by a collection of controllable solutions, where $\varepsilon$ is the control coefficient. On the right, we demonstrate that for each $\varepsilon$, our solution is guaranteed with the Pareto optimality.

In this paper, we focus on the unlearning problem in Image-to-Image (I2I) generative models (Yang et al., 2023), where unlearning is defined by the model's incapacity to reconstruct the full image from a partially cropped one (Li et al., 2024a), as shown in Figure 1. Previous study (Li et al., 2024a) frames machine unlearning in generative models as a single-objective optimization problem, with the loss defined as a combination of performance on both the forget and retain sets. However, this approach faces **three main challenges**: i) First and foremost, this approach offers a fixed result, ignoring the real-world need for flexible trade-offs between model utility and unlearning completeness aligned with varying user expectations. Regrettably, this challenge remains overlooked in the majority of current research on unlearning. ii) This approach relies wholly on fine-tuning with manual terminating conditions, lacking a theoretical guarantee for convergence. iii) This approach integrates two optimization objectives into a single loss function, which compromises unlearning efficiency due to the competition or conflict between different objectives.

To address these challenges, we propose a **controllable unlearning** approach that *provides a set of Pareto optimal solutions to cater to varied user expectations.* Users can select a solution based on the degree of unlearning completeness through a simple control coefficient $\varepsilon$. Specifically, we reframe machine unlearning of I2I generative models into a bi-objective optimization problem (Kim & De Weck, 2005), i.e., unlearning the forget set (1st objective, unlearning completeness) while preserving the retain set (2nd objective, model utility). Due to legislation requirements, the first objective prioritizes the second objective, meaning that minimizing the negative impact on the retain set only arises once the unlearning objective is sufficiently optimized. Therefore, we reformulate the bi-objective optimization problem into a $\varepsilon$-constrained optimization problem, where the unlearning objective is treated as a constraint (primary to satisfy) and $\varepsilon$ is the control coefficient. Utilizing gradient-based methods to solve this $\varepsilon$-constrained optimization, we can obtain two Pareto optimal solutions for the boundaries of unlearning with theoretical guarantee, which can be used to determine the valid range of values for $\varepsilon$. Subsequently, we select the value of $\varepsilon$ within its valid range and relax the constraints on the unlearning objective by increasing $\varepsilon$. As a result, we obtain a set of solutions that dynamically fulfill user's varied expectations regarding the trade-off between unlearning completeness and model utility. Finally, to enhance the efficiency of unlearning, we analyze the convergence rates of our unlearning framework under various settings of the control function which is utilized to govern the direction of parameter updates. The main contributions of this paper are summarized as follows:

- We focus on I2I generative models, and propose a controllable unlearning approach that balances unlearning completeness and model utility, providing a set of solutions to fulfill varied user expectations. To the best of our knowledge, we are the first to study controllable unlearning.

- We reformulate the machine unlearning of generative models as a $\varepsilon$-constrained optimization problem with unlearning the forget set as the constraint, guaranteeing optimal theoretical solutions for the boundaries of unlearning. By progressively relaxing the unlearning constraint, we obtain the Pareto set and plot the corresponding Pareto front.

- We utilize gradient-based methods to solve the $\varepsilon$-constrained optimization problem. To enhance the efficiency of unlearning, we analyze our framework's performance across different settings of the control function and validate with multiple combinations.

- We conduct extensive experiments to evaluate our proposed method over diverse I2I generative models. The results from two large datasets demonstrate that the Pareto optimal solutions yielded by our method significantly outperform baseline methods. Additionally, the solution set achieves controllable unlearning to fulfill varied expectations regarding the trade-off between unlearning completeness and model utility.

# 2 RELATED WORK

## 2.1 I2I GENERATIVE MODELS

Many computer vision tasks can be formulated as I2I generation processes, e.g., style transfer (Zhu et al., 2017), image extension (Chang et al., 2022), restoration (Teterwak et al., 2019), and image synthesis (Yu et al., 2020). There are mainly three architectures for I2I generative models, i.e., Auto-Encoders (AEs) (Alain & Bengio, 2014), Generative Adversarial Networks (GANs) (Goodfellow et al., 2014), and diffusion models (Ho et al., 2020). AEs mainly aim to reduce the mean squared error between generated and ground truth images but often produce lower-quality outputs (Dosovitskiy et al., 2021; Esser et al., 2021). GANs, through adversarial training, significantly improve generation quality, despite their unstable training (Arjovsky et al., 2017; Gulrajani et al., 2017; Brock et al., 2019). Diffusion models, which use a diffusion-then-denoising approach, aim for stable training and high-quality generation by minimizing the distributional distance between generated images and ground truth images (Ho et al., 2020; Song & Ermon, 2020; Salimans & Ho, 2022). However, diffusion models require a greater amount of data and computational resources (Saharia et al., 2022b; Rombach et al., 2022). In this paper, we aim to design a universal unlearning method that can be applied across different I2I models.

## 2.2 MACHINE UNLEARNING

Machine unlearning aims to eliminate the influence of specific training data (unlearning target) from a trained model. A naive approach is to retrain the model from scratch using a modified dataset that excludes the unlearning target. However, this approach can be computationally prohibitive in practice. Based on the degree of unlearning completeness, machine unlearning can be categorized into exact unlearning and approximate unlearning (Xu et al., 2023).

**Exact unlearning** aims to ensure that the unlearning target is fully unlearned, i.e., as complete as retraining from scratch (Bourtoule et al., 2021; Yan et al., 2022; Li et al., 2024b). This approach, which typically relies on retraining, is limited to unlearning specific instances and cannot be readily extended to generative models with strong feature generalizations. **Approximate unlearning** aims to obtain an approximate model, whose performance closely aligns with a retrained model (Golatkar et al., 2020; Sekhari et al., 2021). This approach estimates the influence of unlearning targets, and updates the model accordingly, usually through gradient-based updates, avoiding full retraining (Basu et al., 2021; Li et al., 2023b). However, accurate influence estimation is still challenging (Graves et al., 2021), reducing the applicability of this approach to generative models.

In text-to-image generative models, unlearning typically refers to concept erasure Gandikota et al. (2023); Pham et al. (2023); Lu et al. (2024), such as removing an abstract concept like the artistic style represented by "Van Gogh" or the entity concept represented by "Musk". In contrast, in I2I generative models, the objective of unlearning is to remove the knowledge the model has acquired from a specific set of samples. In this paper, We focus on I2I generative models. The exploration of unlearning is accomplished by minimizing a composite loss, which is a combination of training loss on the retain and the forget sets (Li et al., 2024a). This approach is highly dependent on manual parameter tuning and cannot guarantee unlearning completeness. As for comparison, the solutions yielded by our proposed framework are theoretically guaranteed with Pareto optimality.

# 3 PRELIMINARY

## 3.1 UNLEARNING PRINCIPLES

As outlined in (Chen et al., 2022; Li et al., 2024c), an unlearning task typically has three main principles: i) unlearning completeness, which involves eliminating the influence of specific data

from an already trained model; ii) unlearning efficiency, which focuses on enhancing the speed of the unlearning process; and iii) model utility, which aims to ensure that the performance of the unlearned model remains comparable to that of a model retrained from scratch.

## 3.2 PARETO OPTIMALITY

Given a multi-objective optimization problem, where $f_i(\theta)$ represents the $i$-th objective, the problem can be formalized as: $\min_\theta f(\theta) = (f_1(\theta), f_2(\theta), \cdots, f_m(\theta))^\top$.

**Pareto dominance.** Let $\theta^a$, $\theta^b$ be two points in feasible set $\Omega$, $\theta^a$ is said to dominate $\theta^b$ $(\theta^a \prec \theta^b)$ if and only if $f_i(\theta^a) \leq f_i(\theta^b)$, $\forall i \in \{1, \ldots, m\}$ and $f_j(\theta^a) < f_j(\theta^b)$, $\exists j \in \{1, \ldots, m\}$.

**Pareto optimality (Lin et al., 2019).** A point $\theta^*$ is Pareto optimal if there is no $\hat{\theta} \in \Omega$ for which $\hat{\theta} \prec \theta^*$. The collection of all such Pareto optimal points forms the Pareto set, and the surface of this set in the loss space is called the Pareto front.

## 3.3 I2I GENERATIVE MODEL UNLEARNING

**Model architecture.** Encoder-decoder structures are widely used in I2I models, with: i) an encoder $E_\gamma$ reducing images to the latent space, and ii) a decoder $D_\phi$ reconstructing images from the latent space. For model $I_\theta$ with input image $x$, the output is:

$$I_\theta(x) = D_\phi(E_\gamma(\mathcal{T}(x))), \tag{1}$$

where $\mathcal{T}(x)$ denotes the cropping operation (such as center cropping or random cropping), and $\theta = \{\gamma, \phi\}$ denotes the full parameter set.

**Unlearning objective.** Define the unlearning task for an I2I generative model $I_{\theta_0}$ involving data partitions $D_f$ (forget set) and $D_r$ (retain set). Consider an $I_{\theta_0}$, i.e., the original model, with training data $D = D_f \cup D_r$. Assume that $I_{\theta_0}$ is proficiently trained to generate satisfactory results on both $D_f$ and $D_r$. The objective of unlearning is to obtain an unlearned model $I_\theta$ that cannot generate satisfactory results on $D_f$ (1st objective, unlearning completeness) while maintaining comparable performance on $D_R$ (2nd objective, model utility). Formally,

$$\max_\theta \Big( Div\big(\mathbb{P}(X_f)\|\mathbb{P}(I_\theta(\mathcal{T}(X_f)))\big) \Big), \text{ and } \min_\theta \Big( Div\big(\mathbb{P}(X_r)\|\mathbb{P}(I_\theta(\mathcal{T}(X_r)))\big) \Big), \tag{2}$$

where $X_f$ and $X_r$ are the variables for ground truth images in $D_f$ and $D_r$, $\mathbb{P}(I_\theta(X))$ is the model output distribution for input variable $X$, and $Div(\cdot\|\cdot)$ represents distributional distance, measured by Kullback-Leibler (KL) divergence in this paper.

Following prior work (Kingma et al., 2019; Xia et al., 2022; Wallace et al., 2023), as the model is proficiently trained, we hypothesize that $I_{\theta_0}$ can approximately replicate the distributions over both forget and retain sets (Kingma et al., 2019; Xia et al., 2022; Wallace et al., 2023), i.e., $\mathbb{P}(X_f) \approx \mathbb{P}(I_{\theta_0}(\mathcal{T}(X_f)))$, and $\mathbb{P}(X_r) \approx \mathbb{P}(I_{\theta_0}(\mathcal{T}(X_r)))$. Let $\mathbb{P}_X := \mathbb{P}(I_{\theta_0}(\mathcal{T}(X)))$ and $\mathbb{P}_{\hat{X}} := \mathbb{P}(I_\theta(\mathcal{T}(X)))$. Then, Eq. (2) can be simplified to:

$$\max_\theta Div(\mathbb{P}_{X_f}\|\mathbb{P}_{\hat{X}_f}), \text{ and } \min_\theta Div(\mathbb{P}_{X_r}\|\mathbb{P}_{\hat{X}_r}), \tag{3}$$

where $\mathbb{P}_{X_f}$ and $\mathbb{P}_{\hat{X}_f}$ represent the output distributions of the forget set before and after unlearning respectively. Similarly, $\mathbb{P}_{X_r}$ and $\mathbb{P}_{\hat{X}_r}$ represent those for the retain set.

## 4 METHODOLOGY

In this section, we first introduce a controllable unlearning framework for I2I generative models, which formulates unlearning as a constrained optimization with the unlearning objective as a constraint. We utilize a gradient-based method to obtain the boundaries of unlearning. Then we relax the constraint within the boundaries to derive a set of Pareto optimal solutions to fulfill varied user expectations.

### 4.1 $\varepsilon$-CONSTRAINED OPTIMIZATION FORMULATION

The unlearning task for I2I models is reformulated as a bi-objective optimization (Eq. (3)), with the first objective to maximize $Div(\mathbb{P}_{X_f}||\mathbb{P}_{\hat{X}_f})$. Nonetheless, the value of $Div(\cdot||\cdot)$ can theoretically be maximized to infinity, yielding an infinite number of possible $\mathbb{P}_{\hat{X}_f}$ (Li et al., 2024a), consequently resulting in extremely diminished model utility. To balance unlearning completeness and model utility, we bound $Div(\mathbb{P}_{X_f}||\mathbb{P}_{\hat{X}_f})$ by Lemma 1.

**Lemma 1** (Divergence Upper Bound (Cover & Thomas, 2012)). *Assuming the forget set with distribution $\mathbb{P}_{X_f}$ characterized by a zero-mean and covariance matrix $\Sigma$, and a signal $\mathbb{P}_{\hat{X}_f}$ with the same statistical properties, the maximal KL divergence is realized when $\mathbb{P}_{\hat{X}_f} = \mathcal{N}(0, \Sigma)$.*

$$Div(\mathbb{P}_{X_f}||\mathbb{P}_{\hat{X}_f}) \leq Div(\mathbb{P}_{X_f}||\mathcal{N}(0, \Sigma)). \tag{4}$$

As image normalization typically involves mean subtraction (Elasri et al., 2022), we can assume $\mathbb{P}_{X_f}$ and $\mathbb{P}_{\hat{X}_f}$ follow zero-mean distributions for conciseness without sacrificing generality. Lemma 1 reveals that the upper bound of $Div(\mathbb{P}_{X_f}||\mathbb{P}_{\hat{X}_f})$ is achieved when $\mathbb{P}_{\hat{X}_f} \sim \mathcal{N}(0, \Sigma)$. This suggests that maximizing $Div(\mathbb{P}_{X_f}||\mathbb{P}_{\hat{X}_f})$ equates to minimizing $Div(\mathbb{P}_{\hat{X}_f}||\mathcal{N}(0, \Sigma))$. Consequently, we rewrite Eq. (3) as:

$$\min_{\theta} Div(\mathcal{N}(0, \Sigma)||\mathbb{P}_{\hat{X}_f}), \text{ and } \min_{\theta} Div(\mathbb{P}_{X_r}||\mathbb{P}_{\hat{X}_r}). \tag{5}$$

As both terms in Eq. (5) depend on $\theta$, we define $f_1(\theta) := Div(\mathcal{N}(0, \Sigma)||\mathbb{P}_{\hat{X}_f})$ and $f_2(\theta) := Div(\mathbb{P}_{X_r}||\mathbb{P}_{\hat{X}_r})$ for conciseness. However, unlike classification models where their outputs are precisely univariate discrete distributions (Kurmanji et al., 2024; Zhang et al., 2024a; Shaik et al., 2022; Wu et al., 2024a), high-dimensional KL divergence calculations in I2I generative models are intractable. Thus, following (Li et al., 2024a), we adopt the $L_2$ loss as a surrogate. Due to privacy legal requirements, unlearning objectives typically takes precedence. Thus, we set $f_1(\theta)$ as the primary constraint and treat Eq. (5) as a $\varepsilon$-constrained optimization problem:

$$\min_{\theta \in \mathbb{R}^d} f_2(\theta) \quad \text{s.t.} \quad f_1(\theta) \leq \varepsilon, \tag{6}$$

where $\varepsilon$ is a parameter to control the completeness of unlearning. We minimize $f_2(\theta)$ inside the feasible set $\Omega = \{\theta : f_1(\theta) \leq \varepsilon\}$, which implies that our priority lies in unlearning the forget set rather than mitigating performance degradation on the retain set.

### 4.2 SOLVING THE $\varepsilon$-CONSTRAINT OPTIMIZATION

To solve the $\varepsilon$-constrained optimization problem in Eq. (6), approaches such as Sequential Quadratic Programming (SQP) (Nocedal & Wright; Bonnans et al., 2006), penalty function method (Yeniay, 2005), and interior point method (Renegar, 2001) are commonly employed. Given the extensive parameter set of the I2I generative model, we select a special variant of the SQP algorithm for its lower complexity and comparable convergence guarantee (Nocedal & Wright; Gill & Wong, 2011; Gong et al., 2021).

Specifically, we employ a gradient-based method to solve Eq. (6), updating the parameter by $\theta_{t+1} \leftarrow \theta_t - \mu_t g_t$. Here, $\mu_t > 0$ denotes the step size, and $g_t$ represents the direction of the parameter update, which is determined by solving a convex quadratic programming problem w.r.t. $g$ (for a detailed derivation, please refer to the Appendix C.1):

$$g_t = \min_{g \in \mathbb{R}^d} \left\{ ||\nabla f_2(\theta_t) - g||^2 \quad \text{s.t.} \quad \nabla f_1(\theta_t)^\top g \geq f_1(\theta_t) - \varepsilon \right\}. \tag{7}$$

Due to the inability to obtain the effective range of $\varepsilon$ in the early stages of unlearning, direct computation of $f_1(\theta_t) - \varepsilon$ is not feasible. Consequently, we adjust the constraint of Eq. (7) by employing a control function $\psi(\theta_t)$ (i.e., $\nabla f_1(\theta_t)^\top g \geq \psi(\theta_t)$), which should satisfy $sign(\psi(\theta_t)) = sign(f_1(\theta_t) - \varepsilon)$, where $sign(x) = x/|x|$ for $x \neq 0$ and $sign(0) = 0$. This ensures that the direction of updates remains as consistent as possible before and after the substitution. Further, we provide a summary of our proposed unlearning algorithm in Algorithm 1.

---

**Algorithm 1** $\varepsilon$-Constraint Optimization Algorithm

---

**Require:** Original model $I_{\theta_0}$, forget set $D_f$, retain set $D_r$, control function $\psi(\theta)$, step size $\mu$, co-variance matrix $\Sigma$, numerical stability variable $\varpi = 1e - 7$.

1: **Initial:** Initialize $t = 0$, $I_{\theta_t} = I_{\theta_0}$;
2: **for** $t = 0$ to $T - 1$ **do**
3:     Sample $\{x_f\}$, $\{x_r\}$ and $\{x_n\}$ from $D_f$, $D_r$ and $\mathcal{N}(0, \varepsilon)$ respectively, ensuring that $|\{x_f\}| = |\{x_r\}| = |\{x_n\}|$;
4:     Compute loss:
5:         $f_1(\theta_t) = \|I_{\theta_t}(\mathcal{T}(D_f)) - I_{\theta_0}(\mathcal{T}(x_n))\|_2$
6:         $f_2(\theta_t) = \|I_{\theta_t}(\mathcal{T}(D_r)) - I_{\theta_0}(\mathcal{T}(D_r))\|_2$
7:     Compute gradient: $\nabla f_1(\theta_t)$, $\nabla f_2(\theta_t)$;
8:     Compute the solution to the dual problem of Eq. (7): $\eta_t = \max\left(\frac{\psi(\theta_t) - \nabla f_2(\theta_t)^\top \nabla f_1(\theta_t)}{\|\nabla f_1(\theta_t)\|^2 + \varpi}, 0\right)$;
9:     Compute parameter update direction: $g_t = \nabla f_2(\theta_t) + \eta_t \nabla f_1(\theta_t)$;
10:    Update the parameter of the target model $I_{\theta_{t+1}}$ : $\theta_{t+1} \leftarrow \theta_t - \mu_t g_t$;
11: **end for**
12: **Return** Unlearned model $I_{\theta_T}$;

---

**Assumption 1.** *Assume $f_1(\theta)$ and $f_2(\theta)$ are continuously differentiable, and the trajectory $\{\theta_t : t \in [0, +\infty)\}$ follows the continuous-time dynamics $\dot{\theta}_t = -g_t$, where $g_t$ is defined in Eq. (7) and $\max_{t \in [0, +\infty)} \eta_t < +\infty$.*

The convergence analysis of Algorithm 1 regarding Eq. (6) utilizes the continuous-time framework given by $\dot{\theta}_t = -g_t$, as mentioned in Assumption 1. Please refer to Lemma 3 in Appendix C.2 for further details of convergence.

## 4.3 A CONTROLLABLE UNLEARNING FRAMEWORK

Our controllable unlearning framework consists of two phases. In Phase I, we reformulate Eq. (6) into a special form to obtain the solution for the boundaries of unlearning. In Phase II, we adjust the value $\varepsilon$ within its valid range to relax the unlearning constraint and obtain the Pareto optimal solutions for controllable unlearning. This relaxation of unlearning completeness allows for a controllable trade-off between completeness and model utility, thereby catering to varied user expectations.

**Phase I: Boundaries of unlearning.** The boundaries of unlearning refer to the two Pareto optimal solutions with the highest and lowest degrees of unlearning completeness.

To obtain the Pareto optimal solutions with the highest degrees of unlearning completeness, we reformulate Eq. (6)into the following special form:

$$\min_{\theta \in \mathbb{R}^d} f_2(\theta) \quad \text{s.t.} \quad f_1(\theta) \leq \varepsilon,$$
$$\text{where} \quad \varepsilon = f_1^*, \text{ and } f_1^* := \inf_{\theta \in \mathbb{R}^d} f_1(\theta). \tag{8}$$

The solution of this optimization problem can be obtained by Algorithm 1. According to Assumption 1, we need to ensure that $\psi(\theta) \geq 0$ in Eq. (8) to guarantee the same sign with $f_1(\theta) - \varepsilon$. In this paper, we we simply define $\psi(\theta) = \alpha\|\nabla f_1(\theta)\|^\delta$ with $\alpha > 0$ and $\delta \geq 1$.

**Proposition 1** (Boundary of Pareto Set). *Under Assumption 1, let $f_1^* > -\infty$ and $f_2^* > -\infty$ be the infimum of $f_1(\theta)$, $f_2(\theta)$, respectively. Further, let $\psi(\theta)$ be continuous and $\nabla f_1(\theta)$ be continuously differentiable. If $\theta_t \rightarrow \theta^*$ and $g_t \rightarrow 0$ as $t \rightarrow +\infty$, with $\nabla^2 f_1(\theta)$ of constant rank near $\theta^*$ and $f_1(\theta)$, $f_2(\theta)$ being convex near $\theta_t$, then $\theta^*$ is a Pareto optimal solution and $f_1(\theta^*) = f_1^*$.*

*Proof.* The proof can be found in Appendix C.3. □

Proposition 1 ensures that the solution $\theta_1^*$ obtained by Algorithm 1 for solving Eq. (8) is on the boundary of the Pareto set, specifically refer to the highest degree of unlearning completeness. Meanwhile, $f_1(\theta_1^*)$ achieve the infimum of $f_1(\theta)$.

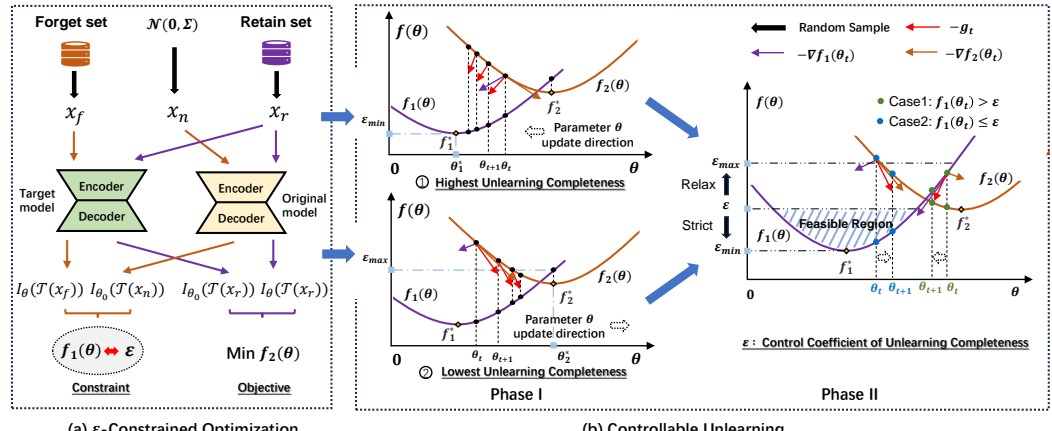

Figure 2: Pipeline of the controllable unlearning framework. (a) shows the unlearning task of the I2I generative model which is framed as a $\varepsilon$-constrained optimization problem. (b) shows that the implementation of controllable unlearning unfolds in two phases: i) initially identifying two boundary points of unlearning, necessitating a strict reduction in $f_1(\theta)$ (or $f_2(\theta)$) for optimality; and ii) then locating the given $\varepsilon$'s Pareto optimal point, with strict reduction in $f_1(\theta)$ when $f_1(\theta_t) > \varepsilon$ and permitting an increase when $f_1(\theta_t) \leq \varepsilon$.

Obtaining the Pareto optimal solution with the lowest unlearning completeness is similar to the process mentioned above, with the difference of exchanging the positions of $f_1(\theta)$ and $f_2(\theta)$ in Eq. (8). This new problem is formulated as $\min_{\theta \in \mathbb{R}^d} f_1(\theta)$, s.t. $f_2(\theta) \leq \varepsilon$, where $\varepsilon = f_2^*$, and $f_2^* := \inf_{\theta \in \mathbb{R}^d} f_2(\theta)$. The solution $\theta_2^*$ obtained by solving this problem is another boundary the Pareto set, i.e., the Pareto optimal solution with the lowest unlearning completeness, with $f_2(\theta_2^*)$ achieving the infimum of $f_2(\theta)$.

**Phase II: Controllable unlearning.** To adjust the trade-off between unlearning completeness and model utility, we relax the unlearning constraint by defining $f_1(\theta_1^*) < \varepsilon < f_1(\theta_2^*)$ in Eq. (6), where $\theta_1^*$ and $\theta_2^*$ have already been obtained in Phase I. Then we rewrite Eq. (8) for controllable unlearning:

$$\min_{\theta \in \mathbb{R}^d} f_2(\theta) \quad \text{s.t.} \quad f_1(\theta) \leq \varepsilon,$$
$$\text{where} \quad \varepsilon > f_1^*, \text{ and } f_1^* := \inf_{\theta \in \mathbb{R}^d} f_1(\theta), \tag{9}$$

where $\varepsilon \in \mathbb{R}$ is used to adjust the completeness of unlearning. In Phase II, according to the sign condition in Assumption 1, we simply set $\psi(\theta) = \beta(f_1(\theta) - \varepsilon)^\delta$ with $\beta > 0, \delta = 2n + 1$ and $n \in \mathbb{N}$.

**Proposition 2** (Interior of Paret Set). *Under Assumption 1, let $f_2^* = \inf_{\theta \in \mathbb{R}^d} f_2(\theta) > -\infty$ and $\sup_{t \in [0, +\infty)} \eta_t = \eta_{max} < +\infty$. If $\theta_t$ is a stationary point with $g_t = 0$ and $\eta_t < +\infty$, and both $f_1(\theta)$ and $f_2(\theta)$ are convex at $\theta_t$, then $\theta_t$ is a Pareto optimal solution w.r.t. $\varepsilon$.*

*Proof.* The proof can be found in Appendix C.4. $\square$

From Proposition 2, Eq. (9) provides a Pareto optimal solution w.r.t. $\varepsilon$. By progressively increasing $\varepsilon$ from $f_1^*$, which is estimated by $f_1(\theta_1^*)$ in Phase I, we can trace a path of Pareto optimal solutions for different completeness of unlearning. As a result, this path offers controllable unlearning for varied user expectations. In addition, our framework demonstrates high unlearning efficiency, with a detailed efficiency analysis provided in Appendix D.

# 5 EXPERIMENTS

## 5.1 EXPERIMENTAL SETTINGS

We evaluate our proposed method on three mainstream I2I generative models, i.e., Masked Autoencoder (MAE) (He et al., 2022), Vector Quantized Generative Adversarial Networks (VQ-GAN) (Li

et al., 2023a), and diffusion probabilistic models (Saharia et al., 2022a). Please refer to Appendix E.1 for the settings of hyperparameters.

**Datasets:** Following (Li et al., 2024a), we conduct experiments on the following two large-scale datasets: i) ImageNet-1K (Deng et al., 2009), from which we randomly select 200 classes, designating 100 of these as the forget set and the remaining 100 as the retain set. Each class contains 150 images, with 100 allocated for training and the remaining for validation; and ii) Places-365 (Zhou et al., 2017), from which we randomly select 100 classes, designating 50 of these as the forget set and the remaining 50 as the retain set. Each class contains 5500 images, with 5000 allocated for training and the remaining 500 for validation.

**Baselines:** We first report the performance of the original model (i.e., before unlearning) as a reference. Following (Li et al., 2024a), we set the following baselines: i) Max Loss (Warnecke et al., 2023; Gandikota et al., 2023), which maximizes the training loss on the forget set; ii) Retain Label (Kong & Chaudhuri, 2023), which minimizes training loss by setting the true values of the retain samples as those of the forget set; iii) Noisy Label (Graves et al., 2021; Gandikota et al., 2023), which minimizes the training loss by introducing Gaussian noise to the ground truth images of the forget set; and iv) Composite Loss (Li et al., 2024a), the State-Of-The-Art (SOTA) method, which builds upon Noisy Label by calculating the loss on the retain set and obtaining their weighted sum, thereby minimizing this weighted training loss.

**Evaluation metrics.** We adopt three different types of metrics to comprehensively compare our method with other baselines: i) Inception Score (IS) of the generated images (Salimans et al., 2016); ii) the Frechét Inception Distance (FID) between the generated images and the ground truth images (Heusel et al., 2017); and iii) the cosine similarity between the CLIP embeddings of the generated images and the ground truth images (Radford et al., 2021). IS evaluates the quality of the generated images independently, while the FID further measures the similarity between the generated and ground truth images. On the other hand, the distance of CLIP embeddings assesses whether the generated images still capture similar semantics. Please refer to Appendix E.2 for more information of evaluation metrics.

## 5.2 UNLEARNING PERFORMANCE

We test our method on image extension, inpainting, and reconstruction tasks. We report the results for center uncropping (i.e., inpainting) in Tabel 1, and the others in Appendix I.1.

**Baseline comparison:** As shown in Table 1, compared to the original model, our method retains almost the same performance on the retain set or only exhibits minor degradation. Meanwhile, there is a significant reduction in the three metrics on the forget set. In contrast, these baselines generally cannot perform well simultaneously on both the forget set and the retain set. For instance, in MAE, Composite Loss has the least performance degradation on the retain set, but its performance on the forget set is also the worst. We also observe similar findings for Max Loss in VQ-GAN. Furthermore, we provide some examples of generated images in Figure 3, and more images in Appendix G.

**T-SNE analysis:** Following (Li et al., 2024a), we conduct a T-SNE analysis (Van der Maaten & Hinton, 2008) to further analyze our method's effectiveness. Using our unlearned model, we generate 50 images for both the retain set and the forget set. We then calculate the CLIP embedding vectors for these images and their corresponding ground truth images. As illustrated in Figure 4, after unlearning, the embeddings of retain set are close to that of the ground truth images, while most of the generated images on the forget set diverge significantly from the ground truth one.

**Unlearning robustness:** We validate the performance of our controllable unlearning framework in different image generation tasks by changing the cropping patterns. The results indicate that our framework is robust to various image generation tasks and generally outperforms baselines, with detailed results provided in Appendix I.1. Moreover, we examine the unlearning effects of our controllable unlearning framework under different crop ratios. The results in Appendix I.3 demonstrate that our framework is robust to different crop ratios. Furthermore, we find that the visual effects of unlearning control are more prominent with larger crop ratios.

**Summary:** These results validate the effectiveness of our proposed method, which is universally applicable to mainstream I2I generative models as well as a variety of image generation tasks, consistently achieving favorable outcomes across all these tasks.

Table 1: Results of center cropping 50% of the images. 'F' and 'R' stand for the forget set and retain set, respectively. Here, "Ours" refers to the boundary points of unlearning obtained in Phase I, that is, the solution with the highest degree of unlearning completeness. The best results are highlighted in **bold**, and secondary results are highlighted with underline.

| | MAE | | | | | | VQ-GAN | | | | | | Diffusion Models | | | | | |
| | IS | | FID | | CLIP | | IS | | FID | | CLIP | | IS | | FID | | CLIP | |
| | F↓ | R↑ | F↑ | R↓ | F↓ | R↑ | F↓ | R↑ | F↑ | R↓ | F↓ | R↑ | F↓ | R↑ | F↑ | R↓ | F↓ | R↑ |
|---|---|---|---|---|---|---|---|---|---|---|---|---|---|---|---|---|---|---|
| Original | 21.59 | 21.83 | 16.28 | 14.87 | 0.88 | 0.88 | 23.74 | 24.06 | 21.80 | 18.17 | 0.78 | 0.85 | 16.90 | 19.65 | 82.12 | 81.51 | 0.89 | 0.91 |
| Max Loss | 15.42 | 16.55 | 129.54 | 87.13 | 0.72 | 0.72 | 19.20 | 21.23 | 23.52 | 43.88 | 0.77 | 0.75 | 17.27 | 18.10 | 95.93 | 108.70 | 0.83 | 0.79 |
| Retain Label | 20.74 | 14.14 | 90.62 | 103.72 | 0.71 | 0.73 | 14.44 | 19.24 | 106.01 | 46.25 | 0.47 | 0.75 | 17.02 | **19.08** | 86.10 | **89.18** | 0.87 | **0.83** |
| Noisy Label | 15.38 | **17.97** | 135.47 | 63.89 | 0.71 | **0.77** | 15.95 | 20.63 | 93.55 | 47.03 | 0.49 | 0.74 | 17.15 | 18.36 | 125.99 | 121.55 | 0.72 | 0.76 |
| Composite Loss | 13.96 | 15.71 | 149.78 | 74.14 | 0.70 | 0.72 | 14.34 | 21.60 | 103.17 | 37.92 | 0.48 | 0.77 | 14.33 | 17.80 | 149.22 | 98.82 | 0.64 | 0.80 |
| **Ours** | **12.33** | 17.47 | **154.60** | 68.453 | **0.69** | 0.75 | **13.23** | **22.55** | **139.21** | **26.39** | **0.46** | **0.82** | **11.84** | 18.47 | **165.05** | 95.42 | **0.55** | 0.81 |

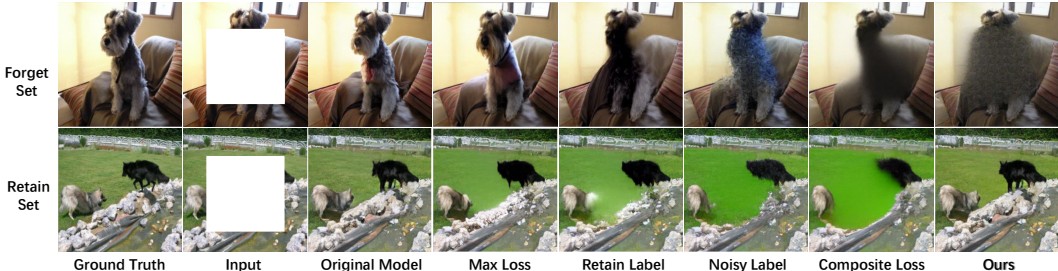

Figure 3: Generated images of cropping 50% at the center of the image on VQ-GAN. From left to right, the images generated by baselines are presented. Our method results in the highest degree of unlearning completeness while maintaining a minimal reduction in model utility.

## 5.3 CONTROLLABLE UNLEARNING

We also evaluate the controllability of our method which provides a set of solutions for varied user expectations. First, we obtain two boundary points of unlearning, thereby establishing the valid range of values for $\varepsilon$. We linearly increase the value of $\varepsilon$ within this range, adding 25% of the range interval each time, to obtain optimum solutions corresponding to different $\varepsilon$ values. We provide some generated images corresponding to these solutions in Figure 1. Due to the space limit, please refer to Appendix H for more examples. For results of more fine-grained control (i.e., smaller increments of the linear increase of $\varepsilon$), please refer to Appendix I.2.

We verify the unlearned models at different $\varepsilon$ values, and report results in Table 2. As $\varepsilon$ increases, we observe a trade-off: the unlearning completeness decreases, while the generated images' performance on the forget set progressively improves, and, simultaneously, the performance on the retain set also improves. This observation clearly demonstrates the controllability of our proposed method, which can cater to varied user expectations. Please refer to Appendix J for additional results of the generated images and T-SNE analysis, which corroborates the above numerical results.

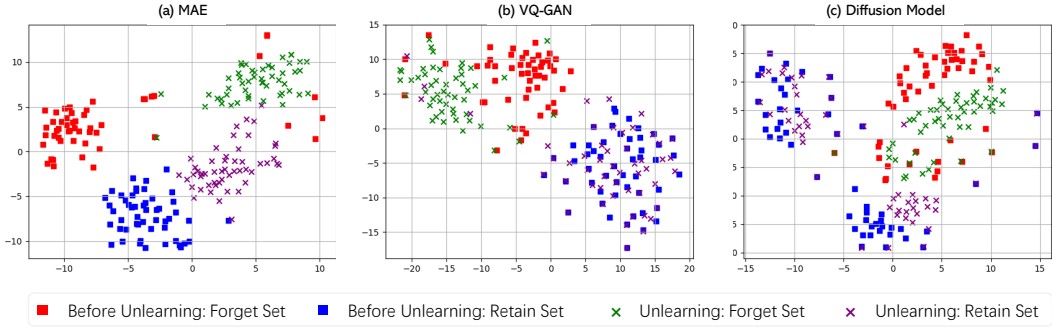

Figure 4: T-SNE analysis between images generated by our method and ground truth images.

Table 2: Results of center cropping 50% of the images under different unlearning completeness. "Highest" and "Lowest" respectively represent the two boundary points of unlearning identified in Phase I. $\varepsilon$ is a coefficient used to control the unlearning completeness in Phase II.

| | MAE | | | | | | VQ-GAN | | | | | | Diffusion Models | | | | | |
| --- | --- | --- | --- | --- | --- | --- | --- | --- | --- | --- | --- | --- | --- | --- | --- | --- | --- | --- |
| | IS | | FID | | CLIP | | IS | | FID | | CLIP | | IS | | FID | | CLIP | |
| | F↓ | R↑ | F↑ | R↓ | F↓ | R↑ | F↓ | R↑ | F↑ | R↓ | F↓ | R↑ | F↓ | R↑ | F↑ | R↓ | F↓ | R↑ |
| Original | 21.59 | 21.83 | 16.28 | 14.87 | 0.88 | 0.88 | 23.74 | 24.06 | 21.80 | 18.17 | 0.78 | 0.85 | 16.90 | 19.65 | 82.12 | 81.51 | 0.89 | 0.91 |
| Highest | 12.33 | 17.47 | 154.60 | 68.453 | 0.69 | 0.75 | 13.23 | 22.55 | 139.21 | 26.39 | 0.46 | 0.82 | 11.84 | 18.47 | 165.05 | 95.42 | 0.55 | 0.81 |
| $\varepsilon$-25% | 17.93 | 20.55 | 85.36 | 59.09 | 0.74 | 0.77 | 14.14 | 22.65 | 130.71 | 24.57 | 0.46 | 0.82 | 15.12 | 19.27 | 137.95 | 84.21 | 0.60 | 0.81 |
| $\varepsilon$-50% | 19.47 | 21.42 | 57.81 | 50.99 | 0.77 | 0.79 | 14.60 | 22.25 | 123.32 | 22.65 | 0.47 | 0.83 | 15.92 | 18.70 | 118.76 | 71.43 | 0.66 | 0.83 |
| $\varepsilon$-75% | 20.68 | 22.87 | 42.51 | 31.80 | 0.80 | 0.82 | 15.20 | 22.53 | 116.59 | 20.63 | 0.47 | 0.84 | 16.33 | 19.53 | 104.21 | 63.62 | 0.73 | 0.83 |
| Lowest | 21.23 | 22.92 | 31.28 | 25.83 | 0.82 | 0.84 | 15.77 | 22.75 | 109.28 | 20.26 | 0.48 | 0.84 | 16.36 | 20.78 | 90.03 | 52.96 | 0.77 | 0.84 |

## 5.4 UNLEARNING EFFICIENCY

To enhance the efficiency of our controllable unlearning framework, we modify the selections of control function $\psi(\theta)$ during various phases. Specifically, we empirically examine the convergence under these conditions to assess the framework's unlearning performance of efficiency. *In Phase I*, with the control function satisfying $\psi(\theta) = \alpha\|\nabla f_1(\theta)\|^{\delta}$, we manipulate the value of the exponent $\delta$ to change the control function. Additionally, we verify the changes in the convergence rates of $f_1(\theta)$ and $f_2(\theta)$ under four different $\delta$ values across three models, with results shown in Appendix K. It is evident that $f_1(\theta)$ and $f_2(\theta)$ achieve an optimal balance in convergence rates when $\delta = 2$, and the overall rate of convergence is fastest. *In Phase II*, where the control function satisfies $\psi(\theta) = \beta(f_1(\theta) - \varepsilon)^{\delta}$, we test the changes in the convergence rates of $f_1(\theta)$ and $f_2(\theta)$ for two different $\delta$ values on three models. To stabilize the optimization process, we scale the form of the control function (i.e., $\psi(\theta) = \beta(f_1(\theta) - \varepsilon)^{\delta}\|\nabla f_1(\theta)\|^2$), selecting two different $\delta$ values, with results presented in Appendix K. It can be observed that at $\delta = 1$ the overall rate of convergence was optimized.

## 6 CONCLUSION

In this paper, we propose a controllable unlearning framework for I2I generative models to overcome the limitation of the existing method's incapability to fulfill varied user expectations. Our approach allows for a controllable trade-off between unlearning completeness and model utility by introducing a control coefficient $\varepsilon$ to control the degrees of unlearning completeness. We reformulate unlearning as a $\varepsilon$-constrained optimization problem and solve it with a gradient-based method to find two boundary points that guide the valid range for $\varepsilon$. Within this range, every chosen value of $\varepsilon$ will lead to a Pareto optimal solution, addressing the existing method's issue of lacking theoretical guarantee. Extensive experiments on two large datasets (i.e., ImageNet-1K and Places-365) across three mainstream I2I models (i.e., MAE, VQ-GAN, diffusion model) demonstrate significant advantages of our method over the SOTA methods with higher unlearning efficiency, and a controllable balance between the unlearning completeness and model utility.

## ACKNOWLEDGMENTS

This work was supported in part by the National Natural Science Foundation of China (No. 62402148) and Ant Group.

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

## A    BROADER IMPACTS AND LIMITATIONS

The abundance of training data not only enhances the performance of generative models but also introduces issues with privacy, unfairness, and bias. Our proposed controllable unlearning framework offers a viable solution to these issues. Our proposed framework is not limited to unlearning in I2I generation models but can be easily extended to other types of generative models, including text-to-image and text-to-text models. However, the unlearning framework presented herein has certain limitations. Note that Propositions 1 and 2 in Section 4 assume the convexity of the objective function and the feasible set. This assumption is essential to guarantee that the yielded solutions are Pareto optimal. In cases where the objective function and the feasible set are non-convex, the solutions obtained from solving Eq. (6) can only be guaranteed to be weakly Pareto optimal (Miettinen, 1999) or Pareto stability (Chen et al., 2024).

## B    DISCUSSION ON THE OBJECTIVE OF UNLEARNING

Describing the unlearning target as inpainting an image using only background content is feasible to some extent, such as concept unlearning (Wu et al., 2024b). For instance, if we aim to protect privacy by unlearning parts of an image generation model that contain personal information (i.e., an abstract concept), we can first identify the region of the image containing such information, then simply mask this region, and subsequently generate a new image through inpainting, ensuring that the model's output aligns with the inpainted new image. However, this approach has two issues:

- Firstly, it must be ensured that the new image generated through inpainting does not contain the information that needs to be forgotten. We believe this can be accomplished by incorporating an additional adversarial discriminator using GAN training strategies or by employing reinforcement strategies.
- Secondly, aligning the model's output with the inpainted new image merely confuses the knowledge learned by the model, increasing uncertainty during generation, which constitutes a superficial form of unlearning. However, based on our experimental experience, if the goal is merely to erase the influence of certain samples on the model, directly aligning with Gaussian noise may yield a more pronounced unlearning effect.

## C    THEORETICAL VALIDATION

### C.1    PROOF OF EQUIVALENCE

Given the original problem

$$\min_{\theta \in \mathbb{R}^d} f_2(\theta) \quad \text{s.t.} \quad f_1(\theta) \leq \varepsilon, \tag{10}$$

which is a constrained nonlinear programming problem. To solve it, we formulate its Lagrangian equation:

$$\mathcal{L}(\theta, \lambda) = f_2(\theta) + \lambda(f_1(\theta) - \varepsilon). \tag{11}$$

Further, we derive the KKT conditions for Eq. (11):

$$\begin{aligned}
&\nabla_\theta \mathcal{L}(\theta^*, \lambda^*) = \nabla f_2(\theta^*) + \lambda^* \nabla [f_1(\theta^*) - \varepsilon] = 0 \\
&f_1(\theta^*) - \varepsilon \leq 0 \\
&\lambda^* \geq 0 \\
&\lambda^*(f_1(\theta^*) - \varepsilon) = 0.
\end{aligned} \tag{12}$$

The standard Newton's Method searches for the solution $\mathcal{L}_\theta(\theta, \lambda) = 0$ by iterating the following equation:

$$\begin{bmatrix} \theta_{t+1} \\ \lambda_{t+1} \end{bmatrix} = \begin{bmatrix} \theta_t \\ \lambda_t \end{bmatrix} - \underbrace{\begin{bmatrix} \nabla_\theta^2 \mathcal{L} & \nabla[f_1(\theta_t) - \varepsilon] \\ \nabla[f_1(\theta_t) - \varepsilon]^T & 0 \end{bmatrix}^{-1}}_{\nabla^2 \mathcal{L}^{-1}} \underbrace{\begin{bmatrix} \nabla_\theta \mathcal{L}(\theta_t, \lambda_t) \\ f_1(\theta_t) - \varepsilon \end{bmatrix}}_{\nabla \mathcal{L}}, \tag{13}$$

where $\nabla_\theta^2$ denotes the Hessian matrix. However, the Newton step $g_t = (\nabla_\theta^2 \mathcal{L})^{-1} \nabla_\theta \mathcal{L}$ cannot be calculated directly and we also have other optimal condition in Eq. (12) introduced by the inequality constraints. Instead, the basic sequential quadratic programming algorithm defines an appropriate search direction $g_t$ at an iterate $(\theta_t, \lambda_t)$, as a solution to the quadratic programming subproblem.

Denoting by $g_t = (g_t^\theta, g_t^\lambda)$ the change in the variables at the current point $(\theta_t, \lambda_t)$, where $(g_t^\theta, g_t^\lambda)$ solve the Newton–KKT system (Nocedal & Wright):

$$
\begin{aligned}
&\nabla_\theta \mathcal{L}(\theta_t, \lambda_t) g_t^\theta + \nabla[f_1(\theta_t) - \varepsilon] g_t^\lambda = -\nabla_\theta \mathcal{L}(\theta_t, \lambda_t) \\
&f_1(\theta_t) - \varepsilon + \nabla[f_1(\theta_t) - \varepsilon] g_t^\theta \leq 0 \\
&\lambda_t + g_t^\lambda \geq 0 \\
&(\lambda_t + g_t^\lambda)\Big( f_1(\theta^*) - \varepsilon + \nabla[f_1(\theta_t) - \varepsilon] g_t^\theta \Big) = 0.
\end{aligned}
\tag{14}
$$

Denoting by $\lambda_{t+1} = \lambda_t + g_t^\lambda$, we have

$$
\begin{aligned}
&\nabla_\theta \mathcal{L}(\theta_t, \lambda_t) g_t^\theta + \nabla[f_1(\theta_t) - \varepsilon] \lambda_{t+1} = -\nabla f_2(\theta_t) \\
&f_1(\theta_t) - \varepsilon + \nabla[f_1(\theta_t) - \varepsilon] g_t^\theta \leq 0 \\
&\lambda_{t+1} \geq 0 \\
&\lambda_{t+1}\Big( f_1(\theta^*) - \varepsilon + \nabla[f_1(\theta_t) - \varepsilon] g_t^\theta \Big) = 0.
\end{aligned}
\tag{15}
$$

It is easy to check that Eq. (15) is the optimality system of the following quadratic problem (QP)

$$
\begin{aligned}
\min_g \quad & f_2(\theta_t) + \nabla f_2(\theta_t)^\top g + \frac{1}{2} g^\top \nabla_\theta^2 \mathcal{L}(\theta_t, \lambda_t) g \\
& f_1(\theta_t) - \varepsilon + \nabla[f_1(\theta_t) - \varepsilon] g \leq 0.
\end{aligned}
\tag{16}
$$

Setting $g_t^\theta = g$, the KKT conditions for Eq. (16) are consistent with the constraints specified in Eq. (15). Further, according to Lemma 2, the optimal solution for Eq. (16), when approaching the optimal solution of the original Problem (i.e., Eq. (10)), satisfies the KKT conditions of Eq. (10). Considering that the models discussed in this paper are all deep neural networks, based on previous studies (Welling & Teh, 2011; Martens, 2016; Zhang et al., 2021; 2022; 2023; 2024b), the initial guess Hessian matrix can be approximated as an identity matrix. Additionally, for consistency with the main text (i.e., $\theta_{t+1} \leftarrow \theta_t - \mu_t g_t$), setting $g = -g_t$ yields the following form:

$$
\begin{aligned}
\min_{g_t} \quad & \nabla f_2(\theta_t)^\top \nabla f_2(\theta_t) - 2\nabla f_2(\theta_t)^\top g_t + g_t^\top g_t \\
& \nabla f_1(\theta_t) g_t \geq f_1(\theta_t) - \varepsilon.
\end{aligned}
\tag{17}
$$

**Lemma 2.** *Theorem of Robinson (1974). Suppose that $\theta^*$ is a local solution of Eq. (10) at which the KKT conditions are satisfied for some $\lambda^*$. Suppose, too, that the linear independence constraint qualification, the strict complementarity condition, and the second-order sufficient conditions hold at $(\theta^*, \lambda^*)$. Then if $(\theta_t, \lambda_t)$ is sufficiently close to $(\theta^*, \lambda^*)$, there is a local solution of the subproblem Eq. (16) whose active set $\mathcal{A}_t$ is the same as the active set $\mathcal{A}(\theta^*)$ of the nonlinear program Eq. (10) at $\theta^*$.*

### C.2 BASIC COMPONENTS

Before exploring the proofs of Propositions 1 and 2, it is essential to define some fundamental concepts and lemmas. This references some works (Boyd & Vandenberghe, 2004; Pardalos et al., 2017; Gong et al., 2021) mentioned earlier; for the sake of readability, we will reiterate them here.

**Penalty Function.** An alternative method to evaluate the optimality of Algorithm 1 involves the $L_1$ penalty function given by:

$$
P_\xi(\theta) = f_2(\theta) + \xi[f_1(\theta) - \varepsilon]_+,
\tag{18}
$$

where $\xi > 0$ is a scaling coefficient. The minima of Eq. (18) align with the solutions to Eq. (6) for sufficiently large values of $\xi$ (Nocedal & Wright).

**First-order KKT Condition and KKT Function.** We revisit the first-order KKT condition (Nocedal & Wright) for the constrained optimization described in Eq. (9). Assume $\theta^*$ is a local optimum

with continuously differentiable $f_1(\theta)$ and $f_2(\theta)$, and $\|\nabla f_1(\theta^*)\| \neq 0$. There exists a Lagrange multiplier $\omega^* \in [0, +\infty)$ such that:

$$\nabla f_2(\theta^*) + \omega^* \nabla f_1(\theta^*) = 0, \quad f_1(\theta^*) \leq \varepsilon, \quad \omega^*(f_1(\theta^*) - \varepsilon) = 0. \tag{19}$$

This setup highlights the importance of $\|\nabla f_1(\theta^*)\| \neq 0$ as a constraint qualification condition.

Utilizing Algorithm 1 for Eq. (9), and for $\eta \geq 0$, the KKT function (Gong et al., 2021) to verify the first-order KKT condition is defined as:

$$K_\tau(\theta_t, \eta_t) = \|\nabla f_2(\theta_t) + \eta_t \nabla f_1(\theta_t)\|^2 + \tau[\psi(\theta_t)]_+ + \eta_t[-\psi(\theta_t)]_+, \tag{20}$$

where $\tau > 0$, and $[x]_+ = \max(x, 0)$. It is clear that $K\tau(\theta_t, \eta_t) \geq 0$ for all $\theta_t \in \mathbb{R}^d$ and $\eta_t \geq 0$, achieving $K_\tau(\theta_t, \eta_t) = 0$ iff $(\theta_t, \eta_t)$ satisfies the first-order KKT condition.

**Second-order KKT Condition and KKT Function**

In the context of Algorithm 1 applied to Eq. (8), we expect that $\|\nabla f_1(\theta_t)\|$ approaches zero, leading to $\eta_t$ potentially diverging to infinity. This scenario indicates a violation of the first-order KKT condition, potentially interpreted as $\eta^* = +\infty$.

While the first-order condition (Eq. (19)) is inadequate, the second-order KKT conditions involving the Hessian $\nabla^2 f_1(\theta)$ are applicable (Dempe et al., 2010). Consider the relaxed form of Eq. (8) as:

$$\min_{\theta \in \mathbb{R}^d} f_2(\theta) \quad \text{s.t.} \quad \nabla f_1(\theta) = 0. \tag{21}$$

If $\theta^*$ is a local minimum of Eq. (8), it coincides with a local minimum of Eq. (21). Assuming $f_2(\theta)$ and $\nabla f_1(\theta)$ are continuously differentiable, with the Hessian $\nabla^2 f_1(\theta)$ maintaining constant rank near $\theta^*$ (Janin, 1984), the first-order KKT condition for Eq. (21) can be formulated. There exists a vector $\omega^* \in \mathbb{R}^d$ such that:

$$\nabla f_2(\theta^*) + \nabla^2 f_1(\theta^*) \omega^* = 0. \tag{22}$$

This condition implies that $\nabla f_2(\theta^*)$ is orthogonal to the null space of $\nabla^2 f_1(\theta^*)$, defining the tangent space of the stationary manifold $\{\theta : \nabla f_1(\theta) = 0\}$ for $f_1(\theta)$.

For verifying local optimality under the constraints of Eq. (8) where $\psi(\theta) \geq 0$, the KKT function is proposed as:

$$K_\tau(\theta_t, \eta_t) = \|\nabla f_2(\theta_t) + \eta_t \nabla f_1(\theta_t)\|^2 + \tau\psi(\theta_t), \tag{23}$$

where $\psi(\theta_t) = 0$ asserts that $\theta_t$ is stationary for $f_1(\theta)$, and $\|\nabla f_2(\theta_t) + \eta_t \nabla f_1(\theta_t)\| = 0$ signifies local optimality with respect to $f_2(\theta)$, aligning with the KKT condition for the relaxed problem $\min_\theta\{f_2(\theta) \text{ s.t. } f_1(\theta) \leq \varepsilon_t\}$, with $\varepsilon_t = f_1(\theta_t)$.

In the analysis of Algorithm 1, a fundamental theorem concerning the behavior of the penalty function $P_\xi(\theta)$ and the KKT function $K_\tau(\theta, \eta)$, given in Eqs. (20) and (23), is essential for understanding the algorithm's convergence and feasibility characteristics. This lemma is stated as follows:

**Lemma 3.** *Theorem 3.2 of Gong et al. (2021). Assume Assumption 1 holds, for any $\xi \geq 0$, we have*

$$\frac{d}{d_t} P_\xi(\theta_t) \leq -K_{\xi-\eta_t}(\theta_t, \eta_t), \forall t \in [0, +\infty). \tag{24}$$

*This equation indicates that $P_\xi(\theta_t)$ is non-increasing w.r.t. time $t$ provided that $K_{\xi-\eta_t}(\theta_t, \eta_t) \geq 0$. This condition is satisfied if $\xi$ is sufficiently large such that $\xi - \eta_t \geq 0$, or when the constraint is met, i.e., $f_1(\theta_t) \leq \varepsilon$, ensuring $[\psi(\theta_t)]_+ = 0$.*

### C.3 PROOF OF PROPOSITION 1

*Proof of Proposition 1.* As $\theta_t$ converges to $\theta^*$ for $t \to +\infty$ and given the continuity of $\psi(\theta)$ and $\nabla f_1(\theta)$, it follows that $\lim_{t\to+\infty} \psi(\theta_t) = \psi(\theta^*)$, and $\lim_{t\to+\infty} \|\nabla f_1(\theta_t)\| = \|\nabla f_1(\theta^*)\|$.

Let $f_1^* = \inf_{\theta \in \mathbb{R}^d} f_1(\theta)$ and $f_2^* = \inf_{\theta \in \mathbb{R}^d} f_2(\theta)$. Since $\psi(\theta) \geq 0$, by substituting Eq. (23) into Eq. (24), we have for any $\xi \geq 0$,

$$\frac{d}{dt}\left(f_2(\theta_t) + \xi[f_1(\theta_t) - \varepsilon]_+\right) \leq -\|\nabla f_2(\theta_t) + \eta_t \nabla f_1(\theta_t)\|^2 - (\xi - \eta_t)\psi(\theta_t), \quad \forall t \in [0, +\infty).$$

Integrating both sides from $0$ to $t$ yields:

$$\int_0^t \left( \left\| \nabla f_2\left(\theta_s\right) + \eta_s \nabla f_1\left(\theta_s\right) \right\|^2 + \left(\xi - \eta_s\right)\psi\left(\theta_s\right)\right) ds \le \left(f_2\left(\theta_0\right) - f_2^*\right) + \xi\left(f_1\left(\theta_0\right) - f_1^*\right).$$
(25)

Given $\psi(\theta) \ge 0$ and $\varepsilon = f_1^*$, Eq. (25) establishes that $\int_0^{+\infty} \psi\left(\theta_t\right) dt \le f_1\left(\theta_0\right) - f_1^* < +\infty$. Consequently, $\lim_{t\to+\infty} \psi\left(\theta_t\right) = \psi\left(\theta^*\right) = 0$.

Given $\theta^*$ as a limit point of $\{\theta_t\}$, there exists an increasing sequence $\{t_n : n = 1, 2, \cdots\}$ such that $t_n \to +\infty$ and $\theta_{t_n} \to \theta^*$ as $n \to +\infty$. The continuity of $\psi(\theta)$ and $\nabla f_1(\theta)$ ensures $\lim_{n\to+\infty} \psi\left(\theta_{t_n}\right) = \psi\left(\theta^*\right) = 0$, and $\lim_{n\to+\infty} \left\|\nabla f_1\left(\theta_{t_n}\right)\right\| = \left\|\nabla f_1\left(\theta^*\right)\right\|$.

Since $\psi\left(\theta^*\right) = 0$ and the sign condition of $\psi(\theta)$, it implies $\mathrm{sign}\left(f_1\left(\theta^*\right) - f_1^*\right) = \mathrm{sign}\left(\psi\left(\theta^*\right)\right) = 0$. Therefore $f_1\left(\theta^*\right) = f_1^*$ and $\theta^*$ is a minimum point of $f_1(\theta)$. This gives $\lim_{n\to+\infty} \left\|\nabla f_1\left(\theta_{t_n}\right)\right\| = \left\|\nabla f_1\left(\theta^*\right)\right\| = 0$.

Given $\lim_{t\to+\infty} g_t = 0$, we deduce that $\lim_{t\to+\infty} \left\|\nabla f_2\left(\theta_t\right) + \eta_t \nabla f_1\left(\theta_t\right)\right\| = \lim_{t\to+\infty} \left\|g_t\right\| = 0$. Additionally, employing $\psi(\theta) \ge 0$, Eq. (23) implies $\lim_{t\to+\infty} K_\tau\left(\theta_t, \eta_t\right) = 0$ for some $\tau > 0$.

Combining $\lim_{t\to+\infty} \left\|\nabla f_2\left(\theta_t\right) + \eta_t \nabla f_1\left(\theta_t\right)\right\| = 0$ and $\nabla f_1\left(\theta^*\right) = \lim_{n\to+\infty} \nabla f_1\left(\theta_{t_n}\right) = 0$, we can derive

$$\left\|\nabla f_2\left(\theta_t\right) + \eta_t \nabla f_1\left(\theta_t\right)\right\| = \left\|\nabla f_2\left(\theta_t\right) + \eta_t\left(\nabla f_1\left(\theta_t\right) - \nabla f_1\left(\theta^*\right)\right)\right\| = \left\|\nabla f_2\left(\theta_t\right) + \nabla^2 f_1\left(\theta_t'\right)\omega_t'\right\|.$$

where $\theta_t'$ is a convex combination of $\theta_t$ and $\theta^*$, and we defined $\omega_t' = \eta_t\left(\theta_t - \theta^*\right)$.

Define $\omega_t = \left(\nabla^2 f_1\left(\theta_t'\right)\right)^+ \nabla f_2\left(\theta_t\right)$, where $\left(\nabla^2 f_1\left(\theta_t'\right)\right)^+$ denotes the Moore-Penrose pseudo-inverse of matrix $\nabla^2 f_1\left(\theta_t'\right)$, which satisfies that

$$\omega_t = \underset{\omega\in\mathbb{R}^d}{\arg\min}\left\{\|\omega\| \quad \text{s.t.} \quad \omega \in \underset{w}{\arg\min}\left\|\nabla f_2\left(\theta_t\right) + \nabla^2 f_1\left(\theta_t'\right)\omega\right\|\right\}.$$

It follows that

$$\left\|\nabla f_2\left(\theta_t\right) + \nabla^2 f_1\left(\theta_t'\right)\omega_t\right\| \le \left\|\nabla f_2\left(\theta_t\right) + \nabla^2 f_1\left(\theta_t'\right)\omega_t'\right\| = \left\|\nabla f_2\left(\theta_t\right) + \eta_t \nabla f_1\left(\theta_t\right)\right\|.$$

Given $\left\|\nabla f_2\left(\theta_{t_n}\right) + \eta_{t_n}\nabla f_1\left(\theta_{t_n}\right)\right\| \to 0$ as $n \to +\infty$, we have $\left\|\nabla f_2\left(\theta_{t_n}\right) + \nabla^2 f_1\left(\theta_{t_n}'\right)\omega_{t_n}\right\| \to 0$. Assuming $\theta_{t_n} \to \theta^*$ and $\theta_{t_n}' \to \theta^*$ as $n \to +\infty$, and by the constant rank condition and relevant corollary of Stewart (1977) (rephrased in Lemma 4), we deduce $\left(\nabla^2 f_1\left(\theta_{t_n}'\right)\right)^+ \to \left(\nabla^2 f_1\left(\theta^*\right)\right)^+$ and hence $\omega_{t_n} \to \omega^*$ as $n \to +\infty$, where $\omega^* := \left(\nabla^2 f_1\left(\theta^*\right)\right)^+ \nabla f_2\left(\theta^*\right)$. Thus, $\left\|\nabla f_2\left(\theta_t\right) + \nabla^2 f_1\left(\theta_t'\right)\omega_t\right\| \to \left\|\nabla f_2\left(\theta^*\right) + \nabla^2 f_1\left(\theta^*\right)\omega^*\right\|$, leading to $\left\|\nabla f_2\left(\theta^*\right) + \nabla^2 f_1\left(\theta^*\right)\omega^*\right\| = 0$, which implies that $\theta^*$ satisfies the second-order KKT conditions for Eq. (22).

Given the convexity of $f_1(\theta)$ and $f_2(\theta)$ with respect to $\theta$, then $f_2(\theta^*)$ is the minimum in the feasible set $\Omega = \{\theta : f_1(\theta) \le \varepsilon\}$, without any $\hat{\theta} \in \Omega$ such that $f_2(\hat{\theta}) < f_2(\theta^*)$. Consequently, $\theta^*$ is a solution to Eq. (8). According to Chankong & Haimes (1982), this solution is unique without further checking, as affirmed by theorem of Miettinen (1999) (rephrased in Lemma 5), $\theta^*$ is Pareto optimal.

Therefore, combining the conclusions, $\theta^*$ is established as both the minimum of $f_1(\theta)$ and Pareto optimal, confirming its status as Pareto optimal for complete unlearning.

$\square$

**Lemma 4.** *Corollary 3.5 of Stewart (1977). Let $\{A_t\}$ be a sequence of matrices converging to $A_*$ as $t \to +\infty$. The condition $\lim_{t\to+\infty} A_t^+ = A_*^+$ is equivalent to the condition that $rank(A_t) = rank(A_*)$ for all $t$ sufficiently large.*

**Lemma 5.** *Theorem 3.2.4 of Miettinen (1999). A point $\theta^* \in \Omega$ is Pareto optimal if it is a unique solution of $\varepsilon$-constraint problem (Eq. (6)) for any given upper bound vector $\varepsilon = \left(\varepsilon_1, \ldots, \varepsilon_{\ell-1}, \varepsilon_{\ell+1}, \ldots, \varepsilon_t\right)^T$.*

## C.4 PROOF OF PROPOSITION 2

*Proof of Proposition 2.* Since $\theta_t$ is stationary, $\dot{\theta}_t = -g_t = 0$, implying $\frac{d}{dt}P_\xi(\theta_t) = 0$ for all $\xi \geq 0$. From Eq. (24), we have $\frac{d}{dt}P_\xi(\theta_t) \leq -K_{\xi-\eta_t}(\theta_t, \eta_t)$. Consequently, $K_{\xi-\eta_t}(\theta_t, \eta_t) \leq 0$ for all $\xi \geq \eta_t$. Setting $\xi = \eta_t + \tau$, where $\tau \geq 0$, it follows that $K_\tau(\theta_t, \eta_t) = 0$. This implies that $\theta^*$ satisfies the first-order KKT conditions for Eq. (19), i.e., there exists a Lagrange multiplier $\eta^* \in [0, +\infty)$ such that

$$\nabla f_2(\theta^*) + \eta^* \nabla f_1(\theta^*) = 0, \quad f_1(\theta^*) \leq \varepsilon, \quad \eta^*(f_1(\theta^*) - \varepsilon) = 0.$$

As affirmed by theorem of Miettinen (1999) (rephrased in Lemma 6), $\theta_t$ is a Pareto optimal solution.

□

**Lemma 6.** *Theorem 3.1.8 of Miettinen (1999). (Karush-Kuhn-Tucker sufficient condition for Pareto optimality) Let the objective and the constraint functions of problem Eq. (9) be convex and continuously differentiable at a decision vector $\theta^* \in \Omega$. A sufficient condition for $\theta^*$ to be Pareto optimal is that there exist multipliers $\boldsymbol{\mu}^* > \mathbf{0}$ and $\boldsymbol{\eta}^* > \mathbf{0}$ such that*

*(1) $\boldsymbol{\mu}^* \nabla f_2(\theta^*) + \boldsymbol{\eta}^* \nabla f_1(\theta^*) = 0$*

*(2) $\boldsymbol{\eta}^*(f_1(\theta^*) - \varepsilon) = 0$.*

## D ENHANCING THE EFFICIENCY OF UNLEARNING

To enhance the efficiency of unlearning, we investigate the influence of the control function $\psi(\theta)$ on convergence rates across different phases, as outlined in the lemma below:

**Lemma 7.** *An extension based on Proposition 3.7 of Gong et al. (2021). Under Assumption 1, with $f_2^* = \inf_{\theta \in \mathbb{R}^d} f_2(\theta) > -\infty$, then: For Phase I, if $\psi(\theta) = \alpha\|\nabla f_1(\theta)\|^\delta$ with $\alpha > 0$ and $\delta \geq 1$, the convergence rates of $f_1(\theta)$ and $f_2(\theta)$ are $O\left(1/t^{\frac{1}{\delta}}\right)$ and $O\left(1/t^{\frac{1}{2}-\frac{1}{2\delta}}\right)$, respectively. For Phase II, if $\psi(\theta) = \beta(f_1(\theta) - \varepsilon)^\delta$ with $\beta > 0$, $\delta = 2n+1$, $n \in \mathbb{N}$, and $\sup_{t \in [0,+\infty)} \eta_t = \eta_{max} < +\infty$, the convergence rate of $[f_1(\theta) - \varepsilon]_+$ is $O\left(1/t^{\frac{1}{\delta}}\right)$.*

Lemma 7 demonstrates that the convergence rate depends on the exponent $\delta$ in $\psi(\theta)$, where higher values of $\delta$ result in a faster convergence rate of $f_1(\theta)$. However, excessively large $\delta$ can also lead to a slower convergence rate of $f_2(\theta)$ and instabilities in training. To balance convergence rate and training stability, we explore various $\varepsilon$ in $\psi(\theta)$ in both phases with extensive empirical studies. The results can be found in Section 5.4.

## E MORE DETAILS OF EXPERIMENTS

### E.1 HYPER-PARAMETER OF EXPERIMENTS

**MAE.** We set the learning rate to $10^{-4}$ with no weight decay. Both baselines and our method employ AdamW as the foundational optimizer with $\beta = (0.90, 0.95)$, with the distinction being that our method necessitates some improvements on the basic optimizer. We set the input image resolution to 224×224 and batch size to 32. Simultaneously, we set the coefficient of $\psi(\theta)$ in Phase I to $\alpha = 5$, and the coefficient of $\psi$ in Phase II to $\beta = 5$, followed by training for 8 epochs. Overall, it takes an hour on an NVIDIA A40 (48G) server.

**VQ-GAN.** We set the learning rate to $10^{-4}$ with no weight decay. Both baselines and our method employ AdamW as the foundational optimizer with $\beta = (0.90, 0.95)$. Our method necessitates some improvements on the basic optimizer. We set the input image resolution to 256×256 and batch size to 16. Simultaneously, we set the coefficient of $\psi(\theta)$ in Phase I to $\alpha = 10$, and the coefficient of $\psi(\theta)$ in Phase II to $\beta = 10$, followed by training for 10 epochs. Overall, it takes two hours on an NVIDIA A40 (48G) server.

**Diffusion model.** We set the learning rate to $10^{-5}$ with no weight decay. Both baselines and our method employ Adam as the foundational optimizer. Our method necessitates some improvements on the basic optimizer. We set the input image resolution to 256×256 and batch size to 16. Simultaneously, we set the coefficient of $\psi(\theta)$ in Phase I to $\alpha = 1$, and the coefficient of $\psi(\theta)$ in Phase II to $\beta = 1$, followed by training for 4 epochs. Overall, it takes twelve hours on an NVIDIA A40 (48G) server.

## E.2 EVALUATION METRICS

**IS.** Following (Li et al., 2024a), for ImageNet-1K, we directly use the Inception-v3 model checkpoint to calculate the IS score. For Places-365, we use the Resnet-50 model checkpoint to calculate IS scores (Zhou et al., 2017).

**FID.** Regardless of whether it is ImageNet-1K or Places-365, we directly use the Inception-v3 model checkpoint to calculate the FID score.

**CLIP.** Following (Li et al., 2024a), whether it is for ImageNet-1K or Places-365, we use the ViT-H-14 model checkpoint to calculate the clip embedding vectors of the generated images and the ground truth images (Radford et al., 2021). Afterward, we calculate the cosine similarity between the two vectors as the clip score.

## F  ROBUSTNESS TO RETAIN SAMPLES AVAILABILITY

In machine unlearning, sometimes the real retain samples are not available due to data retention policies. To tackle this challenge, following (Li et al., 2024a), we assess our method using images from other classes as substitutes for real retain samples. For instance, on ImageNet-1K, since we have already selected 200 classes, we randomly chose some images from the remaining 800 classes to act as a "proxy retain set" during the unlearning process. We incrementally reduce the proportion of real retain samples in the retain set and increased the proportion of proxy retain samples, with the experimental results presented in Table 3. As demonstrated, our method is largely unaffected by the reduced availability of retain samples, indicating robust performance.

Table 3: Results of center cropping 50% of the images under different retain set usage proportions. ↑ indicates higher is better, and ↓ indicates lower is better. 'F' and 'R' stand for the forget set and retain set, respectively. Here, all results are based on the solution with the highest degree of unlearning completeness in Phase I.

| | MAE | | | | | | VQ-GAN | | | | | | Diffusion Models | | | | | |
|---|---|---|---|---|---|---|---|---|---|---|---|---|---|---|---|---|---|---|
| | IS | | FID | | CLIP | | IS | | FID | | CLIP | | IS | | FID | | CLIP | |
| | F↓ | R↑ | F↑ | R↓ | F↓ | R↑ | F↓ | R↑ | F↑ | R↓ | F↓ | R↑ | F↓ | R↑ | F↑ | R↓ | F↓ | R↑ |
| Original | 21.59 | 21.83 | 16.28 | 14.87 | 0.88 | 0.88 | 23.74 | 24.06 | 21.80 | 18.17 | 0.78 | 0.85 | 16.90 | 19.65 | 82.12 | 81.51 | 0.89 | 0.91 |
| 100% | 12.33 | 17.47 | 154.60 | 68.453 | 0.69 | 0.75 | 13.23 | 22.55 | 139.21 | 26.39 | 0.46 | 0.82 | 11.84 | 18.47 | 165.05 | 95.42 | 0.55 | 0.81 |
| 80% | 12.32 | 17.46 | 150.05 | 73.14 | 0.70 | 0.73 | 13.27 | 22.30 | 138.49 | 24.83 | 0.46 | 0.81 | 11.91 | 18.10 | 167.32 | 98.82 | 0.55 | 0.80 |
| 60% | 12.22 | 17.42 | 150.55 | 74.22 | 0.70 | 0.73 | 13.24 | 22.54 | 140.35 | 24.92 | 0.61 | 0.81 | 12.06 | 18.53 | 165.24 | 98.43 | 0.60 | 0.80 |
| 40% | 112.29 | 17.43 | 150.27 | 73.63 | 0.70 | 0.74 | 12.77 | 22.39 | 141.67 | 25.84 | 0.61 | 0.81 | 12.05 | 18.64 | 168.83 | 96.42 | 0.60 | 0.79 |
| 20% | 12.50 | 17.68 | 147.45 | 70.75 | 0.70 | 0.74 | 12.77 | 22.39 | 144.38 | 28.08 | 0.60 | 0.81 | 13.49 | 18.67 | 168.26 | 95.47 | 0.57 | 0.79 |
| 0 | 12.21 | 17.68 | 147.31 | 68.09 | 0.70 | 0.74 | 12.39 | 22.35 | 147.17 | 29.79 | 0.62 | 0.80 | 13.24 | 18.76 | 168.43 | 96.63 | 0.60 | 0.79 |

## G  MORE GENERATED IMAGES: BASELINES VS OURS

We conduct various generative tasks on three mainstream I2I generative models (i.e., MAE, VQ-GAN, and the diffusion model), including image expansion, inpainting, and reconstruction (Zhong & Wang, 2025; Wang et al., 2025; Xiao et al., 2024; Tao et al., 2023), to assess both baselines and our proposed method. Specifically, we conduct evaluations of image inpainting and expansion tasks on VQ-GAN, image reconstruction tasks on MAE, and image inpainting tasks on the diffusion model. The results indicate that our method can adapt to mainstream I2I generative models and various image generation tasks.

**VQ-GAN.** We conduct experiments on image inpainting and expansion task unlearning on VQ-GAN, where examples of the image inpainting tasks are illustrated in Figure 5, and examples of image expansion can be referred to in Appendix I. Our unlearning method is effective for both image inpainting and image expansion tasks, and it significantly surpasses baselines.

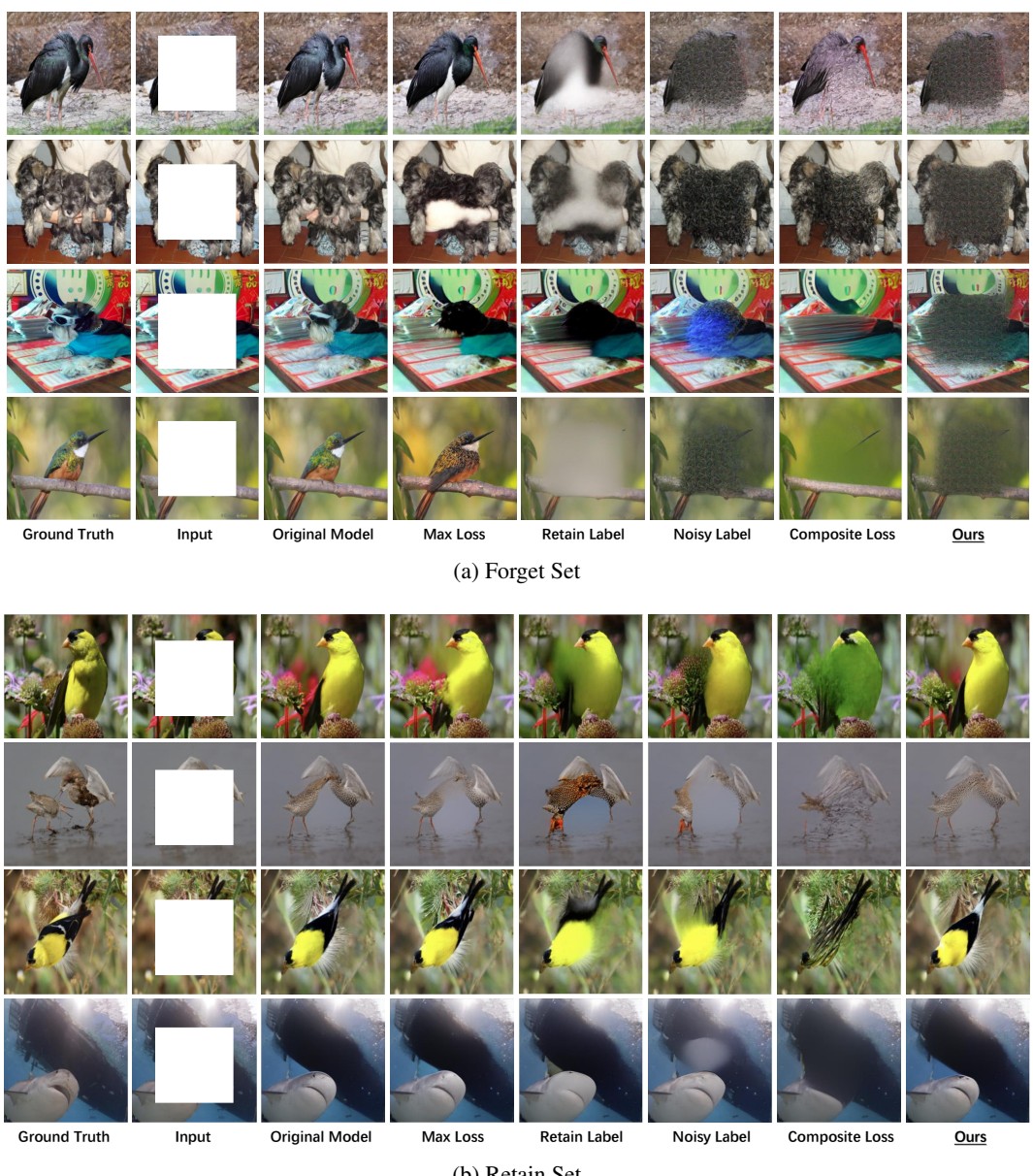

Figure 5: VQ-GAN: generated images of cropping 50% at the center of the image. The upper part (a) represents the forget set, while the lower part (b) represents the retain set. "Ours" denotes the boundary condition of unlearning obtained in Phase I, which represents the point of the highest degree of unlearning completeness. It is evident that our method significantly outperforms baselines in terms of the unlearning effect on the forget set, most closely approximating Gaussian noise, and exhibits the least performance degradation on the retain set.

**MAE.** We conduct experiments on unlearning image reconstruction tasks on the MAE. As shown in figure 6, our unlearning method is also effective in the task of image reconstruction, with the effects of unlearning showing a significant advantage over baselines.

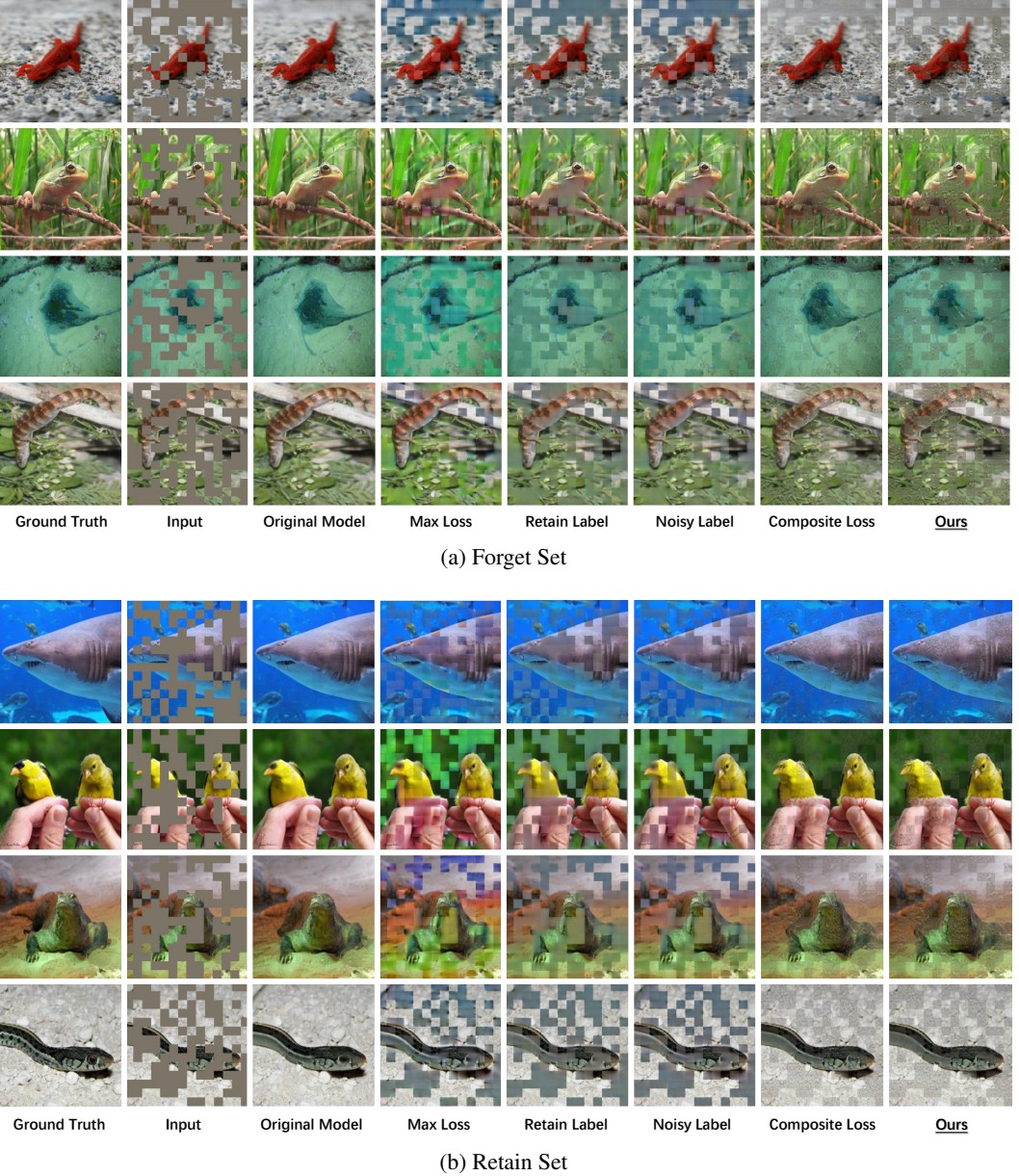

Figure 6: MAE: reconstruction of random masked images. We set the proportion of the random mask to 50%. The upper part (a) represents the forget set, while the lower part (b) represents the retain set. "Ours" denotes the boundary condition of unlearning obtained in Phase I, which represents the point of the highest degree of unlearning completeness.

**Diffusion model.** We validate our unlearning framework on the diffusion model task for image inpainting. As shown in figure 7, the results indicate that our method is equally applicable to diffusion models, and the effectiveness of unlearning surpasses that of baselines.

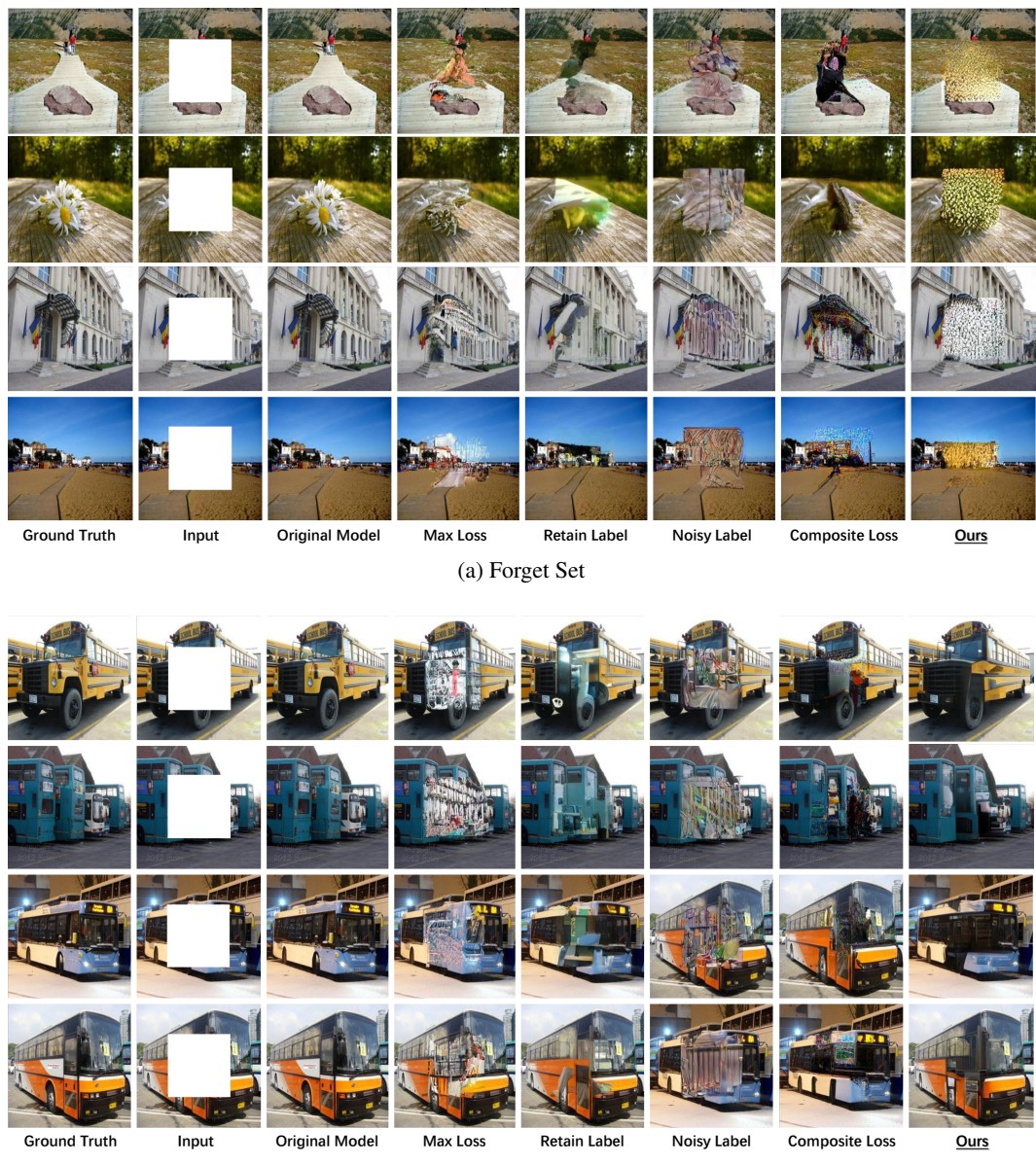

Figure 7: Diffusion model: generated images of cropping 50% at the center of the image. The upper part (a) represents the forget set, while the lower part (b) represents the retain set. "Ours" denotes the boundary condition of unlearning obtained in Phase I, which represents the point of the highest degree of unlearning completeness.

# H MORE GENERATED IMAGES: DIFFERENT DEGREES OF COMPLETENESS

We validate the control effect of our controllable unlearning framework across multiple generative tasks in three mainstream I2I generative models. The results demonstrate that our controllable unlearning framework can effectively control unlearning across various image generation tasks of mainstream I2I generative models.

**VQ-GAN.** We center-cropp the image by 50% and utilize the VQ-GAN for image inpainting. Subsequently, we applied our unlearning framework to enforce unlearning. The results in Figure 8 demonstrate the effectiveness of our method, with the control effect being very pronounced.

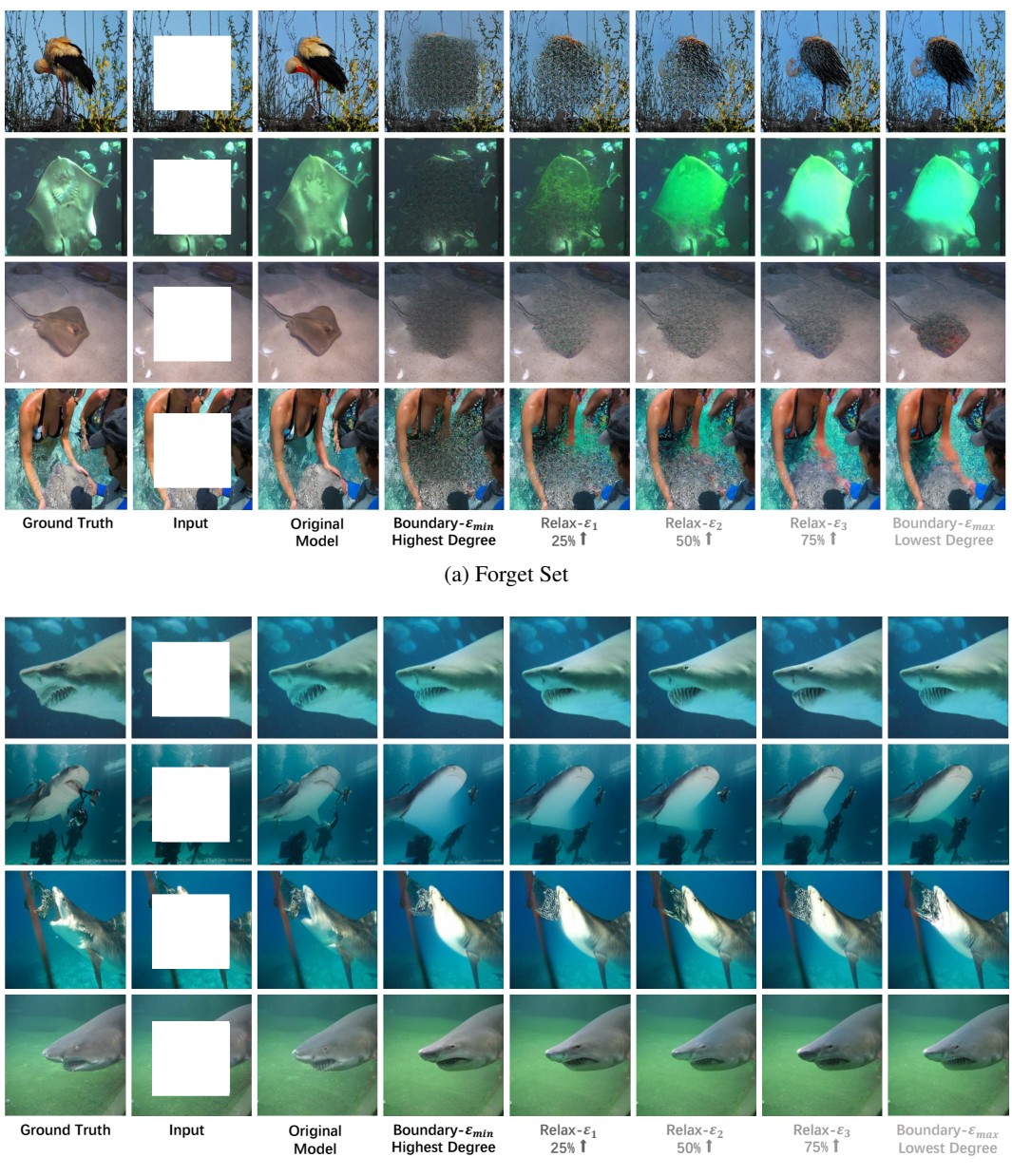

Figure 8: VQ-GAN: generated images of cropping 50% at the center of the image under different degrees of unlearning completeness requirements. The upper half (a) represents the forget set, and the lower half (b) represents the retain set. Our method first determines the two boundary conditions of unlearning, and then linearly increases the value of $\varepsilon$ within its range (here, we increase by 25% each time) to adjust the balance between unlearning completeness and model utility.

**MAE.** We verify the control effect of our controllable unlearning framework within the reconstruction task using the MAE. The results in Figure 9 indicate that our method can effectively control the completeness of unlearning in image reconstruction tasks as well.

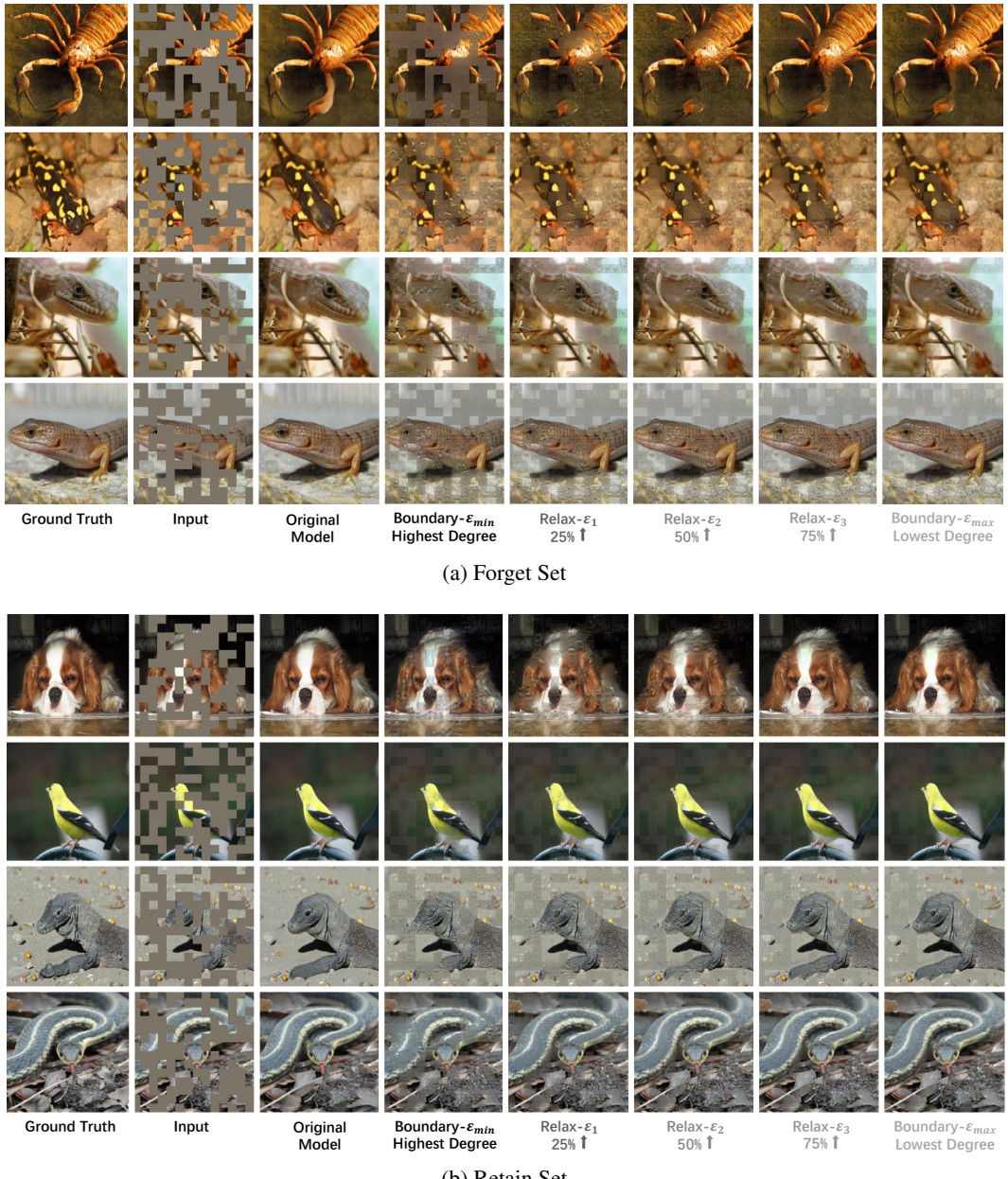

Figure 9: MAE: construction of random masked images under different degrees of unlearning completeness requirements. We set the proportion of the random mask to 50%. The upper half (a) represents the forget set, and the lower half (b) represents the retain set. Our method first determines the two boundary conditions of unlearning, and then linearly increases the value of $\varepsilon$ within its range (here, we increase by 25% each time) to adjust the balance between unlearning completeness and model utility.

**Diffusion model.** We validate the control effect of our controllable unlearning framework within the inpainting task of a diffusion model. As shown in Figure 10, the findings illustrate that our method can effectively adjust the balance between the completeness of unlearning and the utility of the model in the context of a diffusion model.

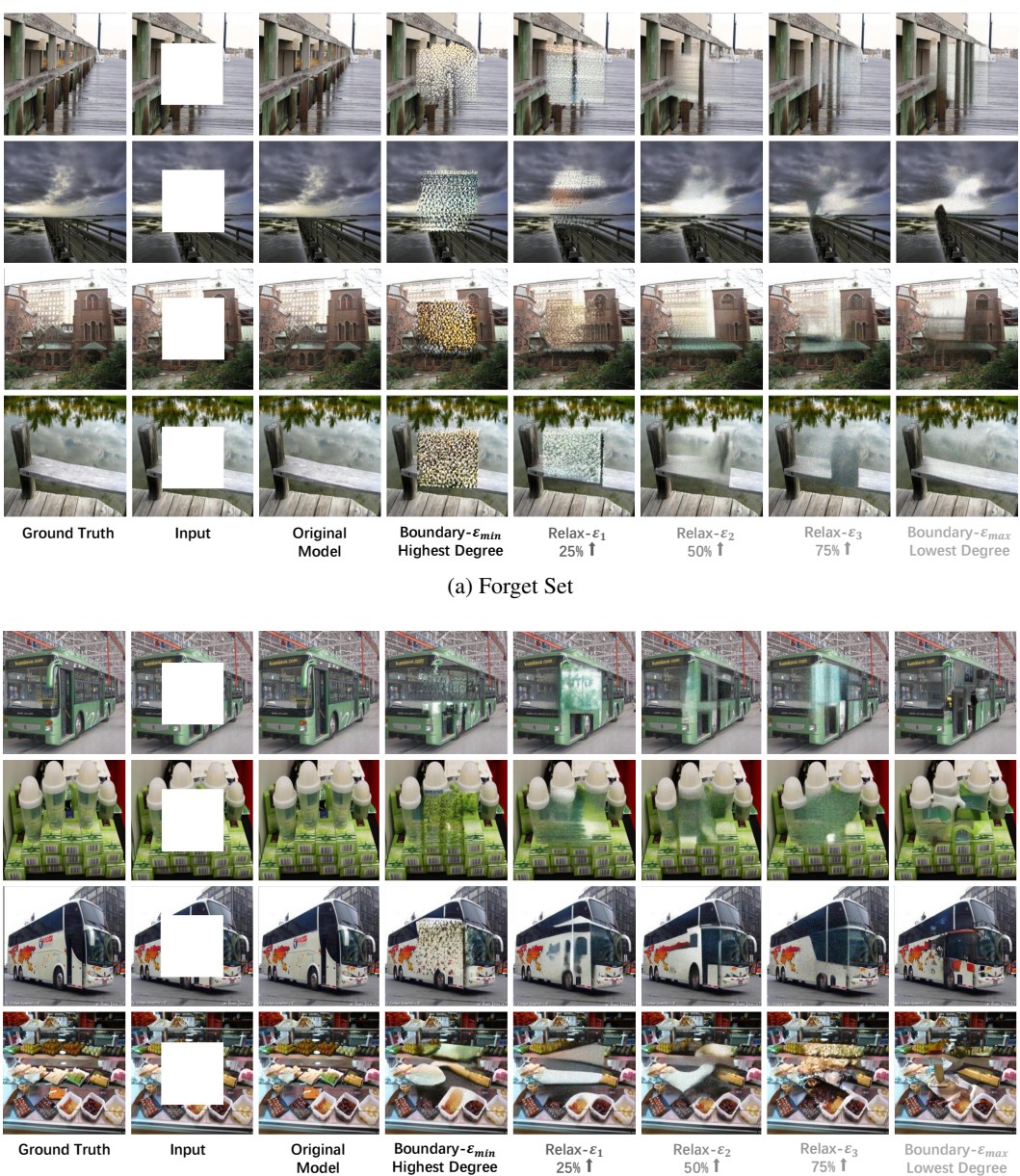

Figure 10: Diffusion model: generated images of cropping 50% at the center of the image under different degrees of unlearning completeness requirements. The upper half (a) represents the forget set, and the lower half (b) represents the retain set. Our method is also effective when applied to the diffusion model.

# I ABLATION STUDY

To verify the robustness of our method on mainstream I2I generative models and various image generation tasks, we conducted the following ablation studies: i) we vary the cropping patterns to

demonstrate robustness across multiple image generation tasks; ii) we decrease the linear increment size of $\varepsilon$ to validate that our method allows for more fine-grained control; and iii) we alter the cropping ratios to confirm the robustness of our method to changes in crop ratio.

## I.1 MORE GENERATIVE TASKS

Similar to validating unlearning in classification models through Membership Inference Attacks Choi & Na (2023), generative models can also be assessed for unlearning robustness by employing attack methods to reconstruct the forget set. Although there is substantial research in this area Kumari et al. (2023); Petsiuk & Saenko (2025), it typically focuses on concept unlearning in text-to-image generative models. In contrast, our focus is on unlearning in image-to-image generative models. Unlike unlearning a single concept, our goal is to unlearn the influence of a set of samples or their distribution on the model. This makes it challenging to validate the effectiveness and robustness of our method through attacks. Specifically, we validate the effectiveness and robustness of our controllable unlearning framework for image extension tasks on VQ-GAN by varying the patterns of cropping. The results indicate that our controllable unlearning framework is robust to different cropping patterns.

### I.1.1 OUTPAINTING TASK

We retain 25% of the image center and utilize VQ-GAN for image outpainting. As shown in Figure 11, our method produces outpainting on the forget set that is most similar to Gaussian noise, and the outpainting performance on the retain set shows the least decline compared to the original model.

### I.1.2 UPWARD EXTENSION TASK

We crop the upper half of the image, retain the lower half, and employ VQ-GAN for image extension. The results in Figure 12 indicate that our method produces extension on the unlearning set that closely resembles Gaussian noise, and on the retain set, the extension performance decreases the least compared to the original model.

### I.1.3 LEFTWARD EXTENSION TASK

We crop the right half of the image, retain the left half, and use VQ-GAN for image extension. As shown in Figure 13, our method produces leftward extension on the forget set that closest resembles Gaussian noise and, on the retain set, the leftward extension performance exhibits the minimal decrease compared to the original model.

## I.2 MORE FINE-GRAINED CONTROL OF UNLEARNING COMPLETENESS

After obtaining two boundary points of unlearning, our controllable unlearning framework linearly increases within its valid range to balance the completeness of unlearning and the utility of the model. However, in the main paper, the increase of $\varepsilon$ is by 25% each time. For example, if the range of $\varepsilon$ is [1,9], then the sequence of $\varepsilon$ values would be {3,5,7}. It is evident that the increments of $\varepsilon$ are quite substantial, which results in a coarser granularity of control. Here, we reduce the linear increment of $\varepsilon$ to extend the effectiveness of our controllable unlearning framework across various image generation tasks in VQ-GAN. The results show that our framework can achieve fine-grained control.

### I.2.1 OUTPAINTING TASK

We retain the central 25% of the image and utilize VQ-GAN for image outpainting. The results in Figure 14 show that the performance of our controllable unlearning framework on the forget set gradually improves with the increase of $\varepsilon$, and the extent of decline in outpainting performance on the retain set, compared to the original model, is also reducing.

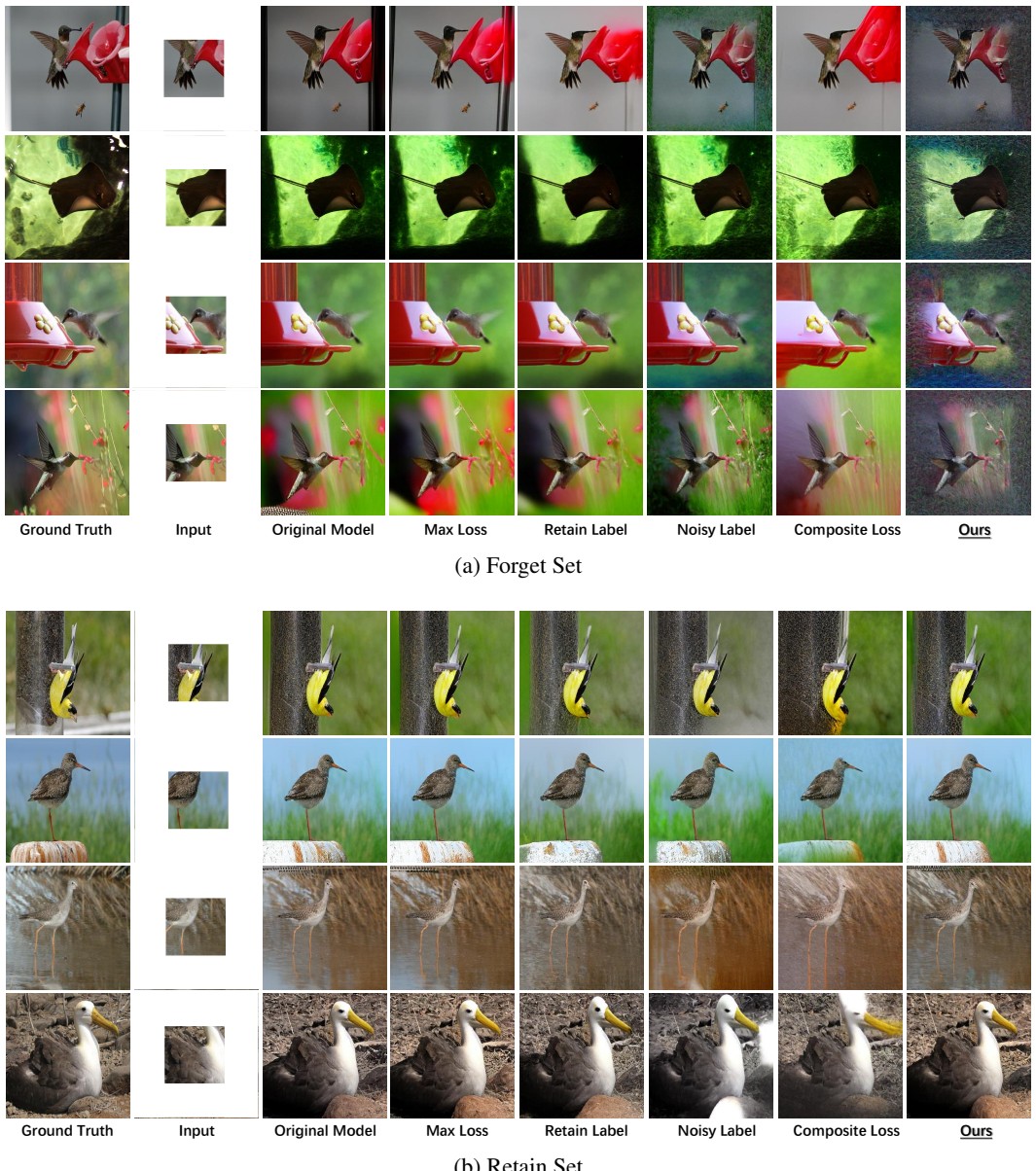

Figure 11: Outpainting by VQ-GAN. We retain 25% of the image center. The upper half (a) designated as the unlearning set and the lower half (b) as the retain set. For each subset, we compared the performance of both the baselines and our method on the outpainting task, where "Ours" represents the boundary condition of unlearning in Phase I, indicating the point of highest degree of unlearning completeness. The results show that our method significantly outperforms the baselines on the outpainting task.

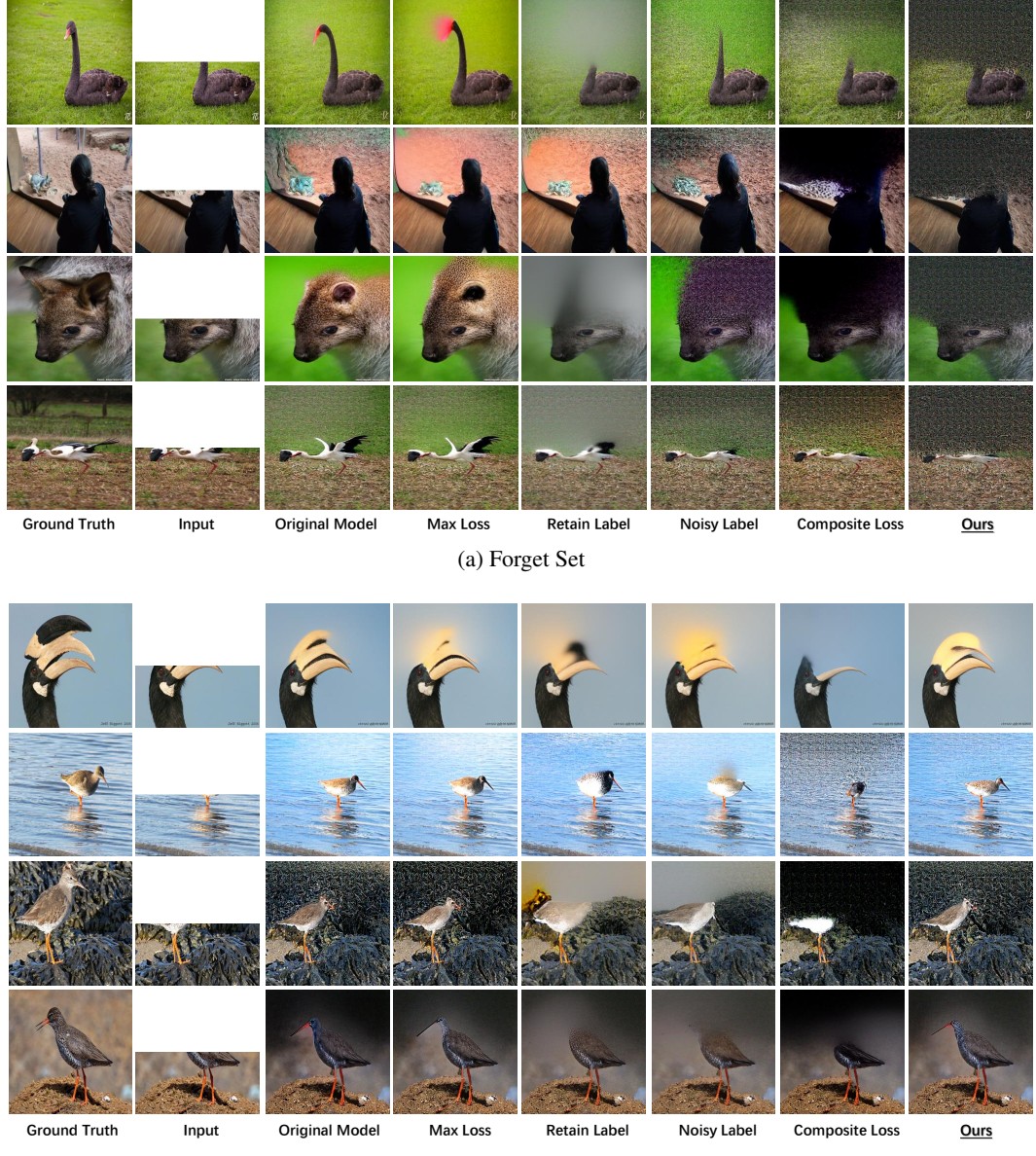

Figure 12: Upward extension by VQ-GAN. We retain 50% of the lower half of the image. The upper half (a) is the forget set, and the lower half (b) is the retain set. For each set, we compare the performance of the baselines and our method on the upward extension task, where "Ours" represents the unlearning boundary condition in Phase I, which is the point of the highest degree of unlearning completeness. The results suggest that our method also significantly outperforms the baselines on the upward extension task.

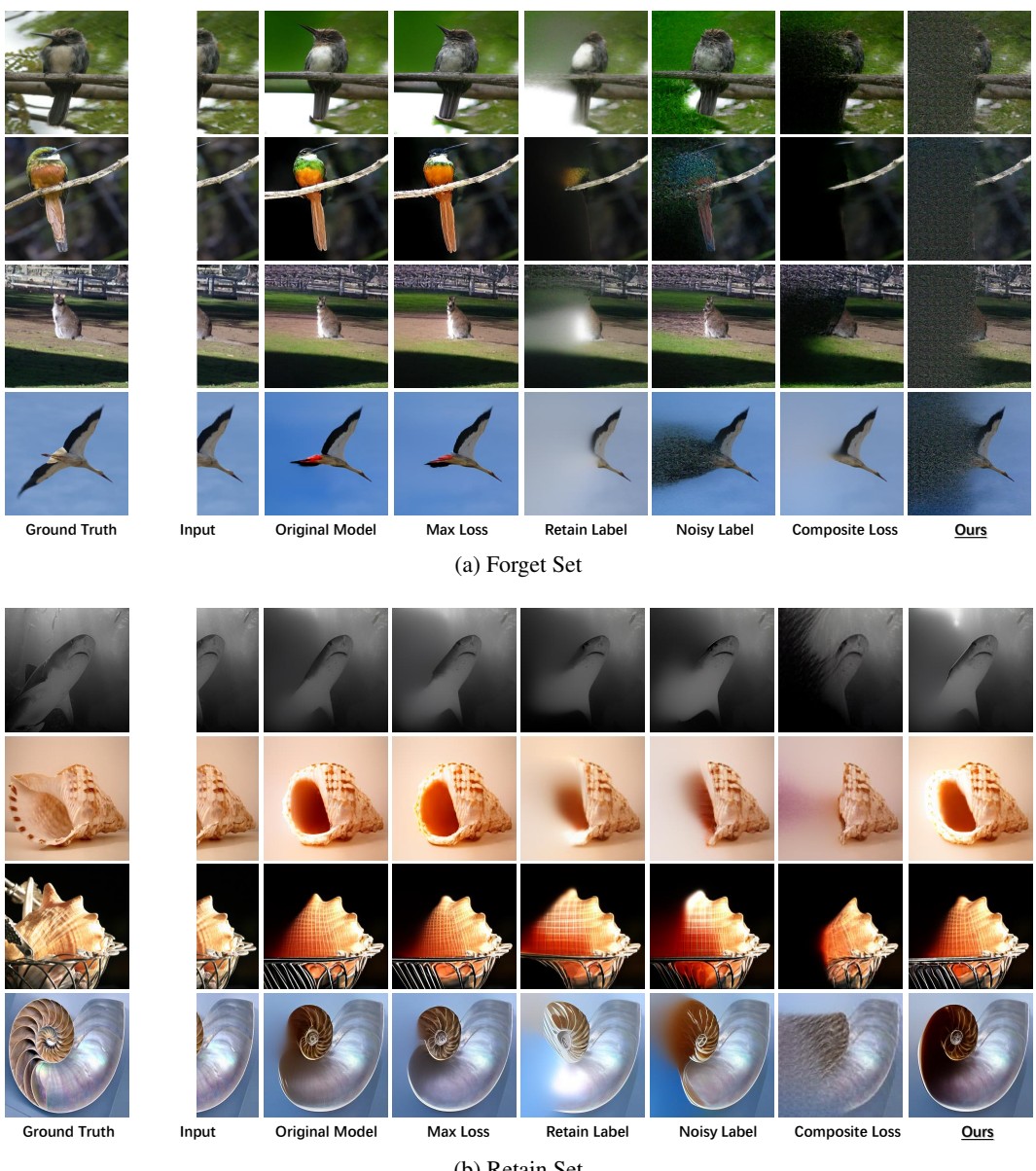

Figure 13: Leftward extension by VQ-GAN. We retain 50% of the right half of the image. The upper half (a) is the forget set, and the lower half (b) is the retain set. For each set, we compare the performance of the baselines and our method on the upward extension task, where "Ours" represents the unlearning boundary condition in Phase I, which is the point of highest degree of unlearning completeness. The results suggest that our method also significantly outperforms the baselines on the upward extension task.

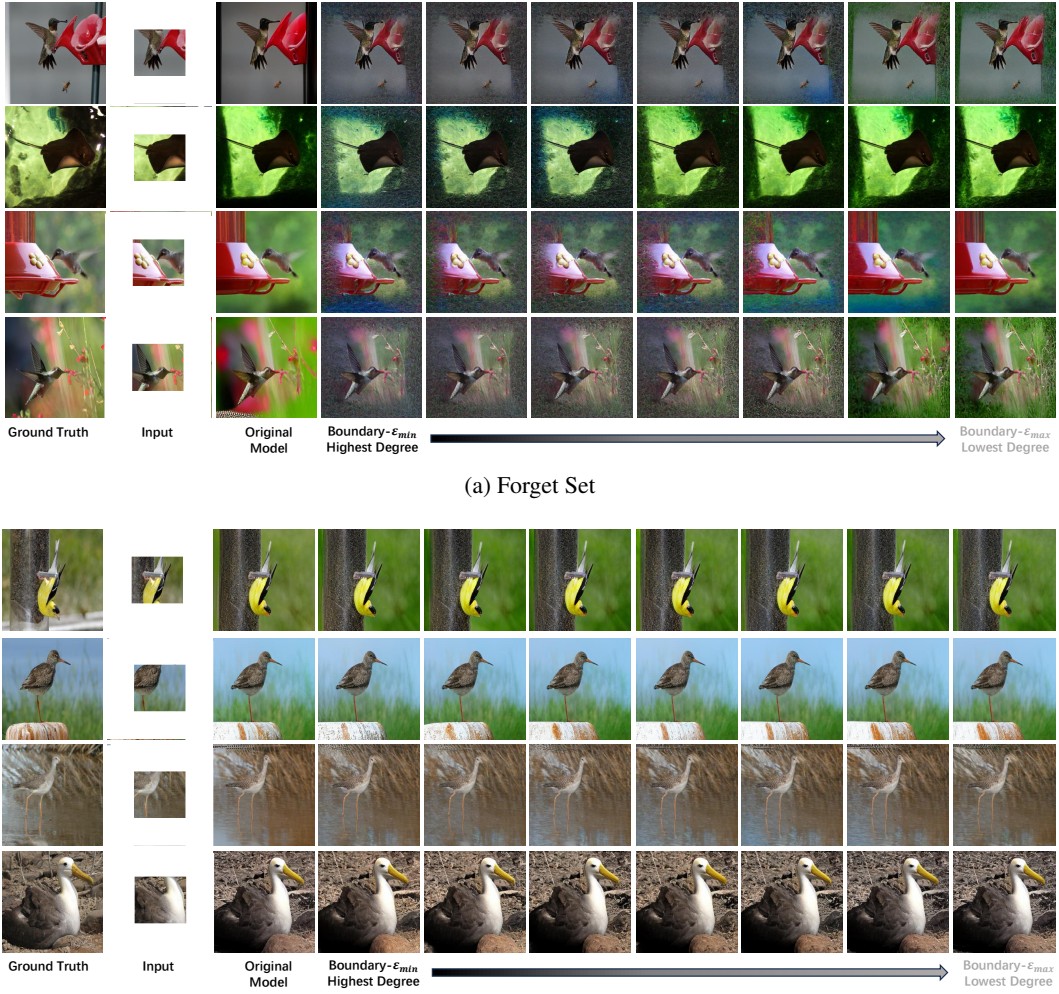

Figure 14: Outpainting by VQ-GAN under different degrees of unlearning completeness. We retain 25% of the image center. The upper half (a) is the forget set, while the lower half (b) is the retain set. For each part, we compare the unlearning effects of our method at different values of $\varepsilon$. "Highest" and "Lowest" represent the conditions of the highest and lowest degree of unlearning completeness, respectively. We increase $\varepsilon$ 16% each time.

### I.2.2 UPWARD EXTENSION TASK

We retain the lower half of the image center and crop the upper half, employing VQ-GAN for image extension. As shown in Figure 15, results indicate that, with an increase in the value of $\varepsilon$, the upward extension effectiveness on the forget set of our controllable unlearning framework gradually improves. Concurrently, the degree of decrease in upward extension effectiveness on the retain set, in comparison to the original model, also diminishes.

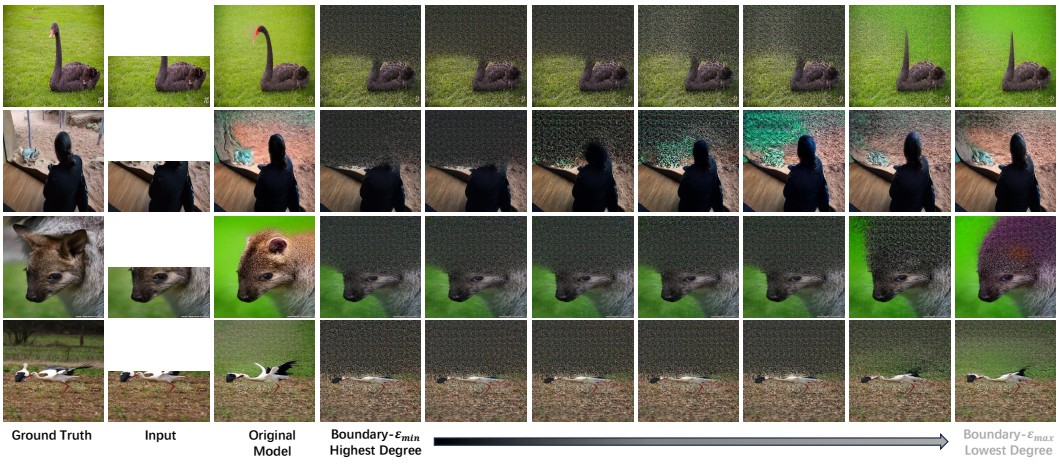

(a) Forget Set

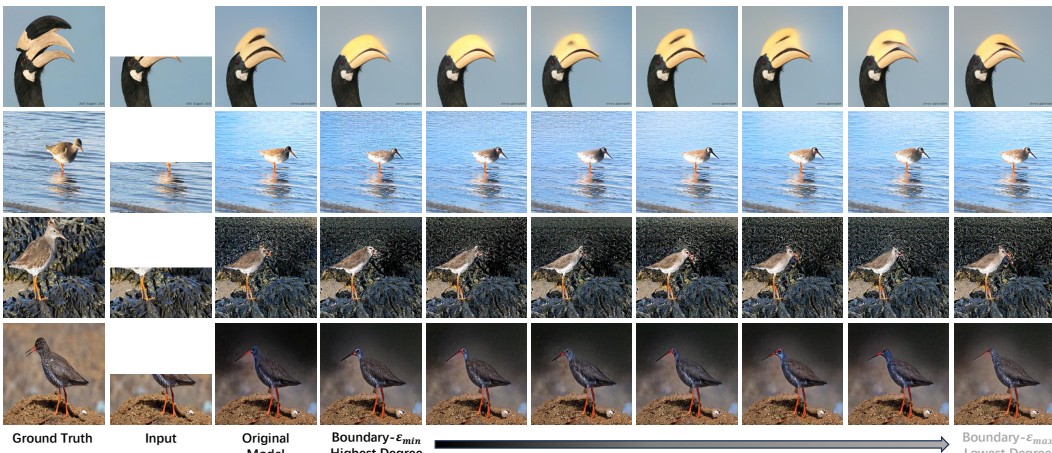

(b) Retain Set

Figure 15: Upward extension by VQ-GAN under different degrees of unlearning completeness. We retain 50% of the lower half of the image. The upper half (a) is the forget set, while the lower half (b) is the retain set. For each part, we compare the unlearning effects of our method at different values of $\varepsilon$. "Highest" and "Lowest" represent the conditions of the highest and lowest degree of unlearning completeness, respectively. We increase $\varepsilon$ 16% each time.

### I.2.3 LEFTWARD EXTENSION TASK

We retain the right half of the image and utilize VQ-GAN to extend the image from the left. The results in Figure 16 demonstrate that the leftward extension performance on the forget set of our controllable unlearning framework progressively improves with the increase of $\varepsilon$, and the reduction in leftward extension performance on the retain set is also diminishing compared to the original model.

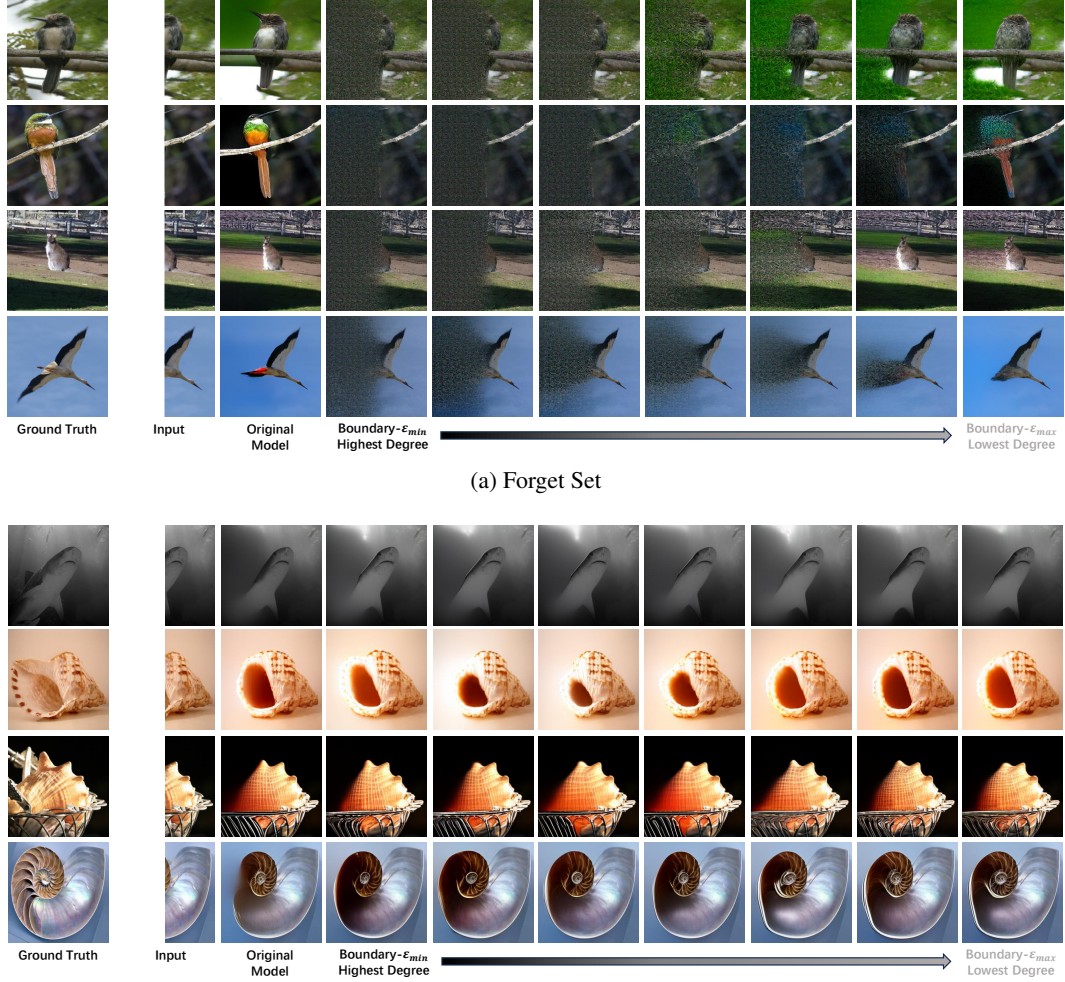

Figure 16: Leftward extension by VQ-GAN under different degrees of unlearning completeness. We retain 50% of the right half of the image. The upper half (a) is the forget set, while the lower half (b) is the retain set. For each part, we compare the unlearning effects of our method at different values of $\varepsilon$. "Highest" and "Lowest" represent the conditions of the highest and lowest degree of unlearning completeness, respectively. We increase $\varepsilon$ 16% each time.

### I.3 Varying Cropping Patterns and Ratios

In the preceding sections, we have demonstrated the performance of our controllable unlearning framework under various cropping patterns, yet the cropping ratio remained constant. By altering the cropping ratio on VQ-GAN, we validate the effectiveness of our controllable unlearning framework at different cropping ratios. The results indicate that our controllable unlearning framework is robust to different cropping ratios. Simultaneously, compared to larger cropping ratios, the extent of variation in the images generated under our controllable unlearning framework will be smaller for smaller cropping ratios.

#### I.3.1 Inpainting Task

We retain one-sixteenth of the image center and use VQ-GAN for image inpainting. The results in Figure 17 show that our controllable unlearning framework significantly outperforms the baselines in terms of unlearning effect on the forget set, most closely approximating Gaussian noise, and exhibits a lesser decline in unlearning effect on the retain set than the baselines. Simultaneously, we can finely control the balance between unlearning completeness and model utility.

#### I.3.2 Downward Extension Task

We crop the bottom 25% of the image and utilize VQ-GAN for image extension from the bottom. As shown in Figure 19, the results demonstrate that our controllable unlearning framework significantly surpasses the baselines in terms of the unlearning effect on the forget set, closely approximating Gaussian noise, and shows a lesser reduction in unlearning effect on the retain set compared to the baselines. At the same time, we can finely adjust the balance between unlearning completeness and model utility.

#### I.3.3 Rightward Extension Task

We crop the right 25% of the image and utilize VQ-GAN for image extension from the bottom. The results in Figure 21 demonstrate that our controllable unlearning framework significantly surpasses the baselines in terms of the unlearning effect on the forget set, closely approximating Gaussian noise, and shows a lesser reduction in unlearning effect on the retain set compared to the baselines. At the same time, we can finely adjust the balance between unlearning completeness and model utility.

## J T-SNE Analysis for Controllable Unlearning

In Table 2 of the main paper, we present the evaluation metrics corresponding to different degrees of unlearning completeness solutions (i.e., IS, FID and CLIP) obtained by our controllable unlearning framework in mainstream I2I generative models. Here, we analyze the images generated at different degrees of unlearning completeness for each corresponding model. We use T-SNE analysis to compare the clip embedding distances between the images generated on the forget set and retain set and the ground truth images. As shown in Figure 23, for any model, under the highest degree of unlearning completeness, the distance between the clip embeddings of the images generated on the forget set by the unlearned model and the ground truth images is larger, while the distance on the retain set is smaller. Simultaneously, as $\varepsilon$ increases, the distance between the clip embeddings of the images generated on the forget set by the unlearning model and the ground truth images gradually decreases (still significantly higher than the situation of the retain set), and the distance on the retain set also gradually decreases. Lastly, among these three mainstream I2I generation model structures, the effect of VQ-GAN is the most significant.

## K Efficiency Experiments for Controllable Unlearning Framework

In the main paper, we analyze the convergence efficiency corresponding to different control functions $\psi(\theta)$ at each phase from a theoretical perspective, and based upon this analysis, we aim to enhance the unlearning efficiency of our controllable unlearning framework. Here, we validate our

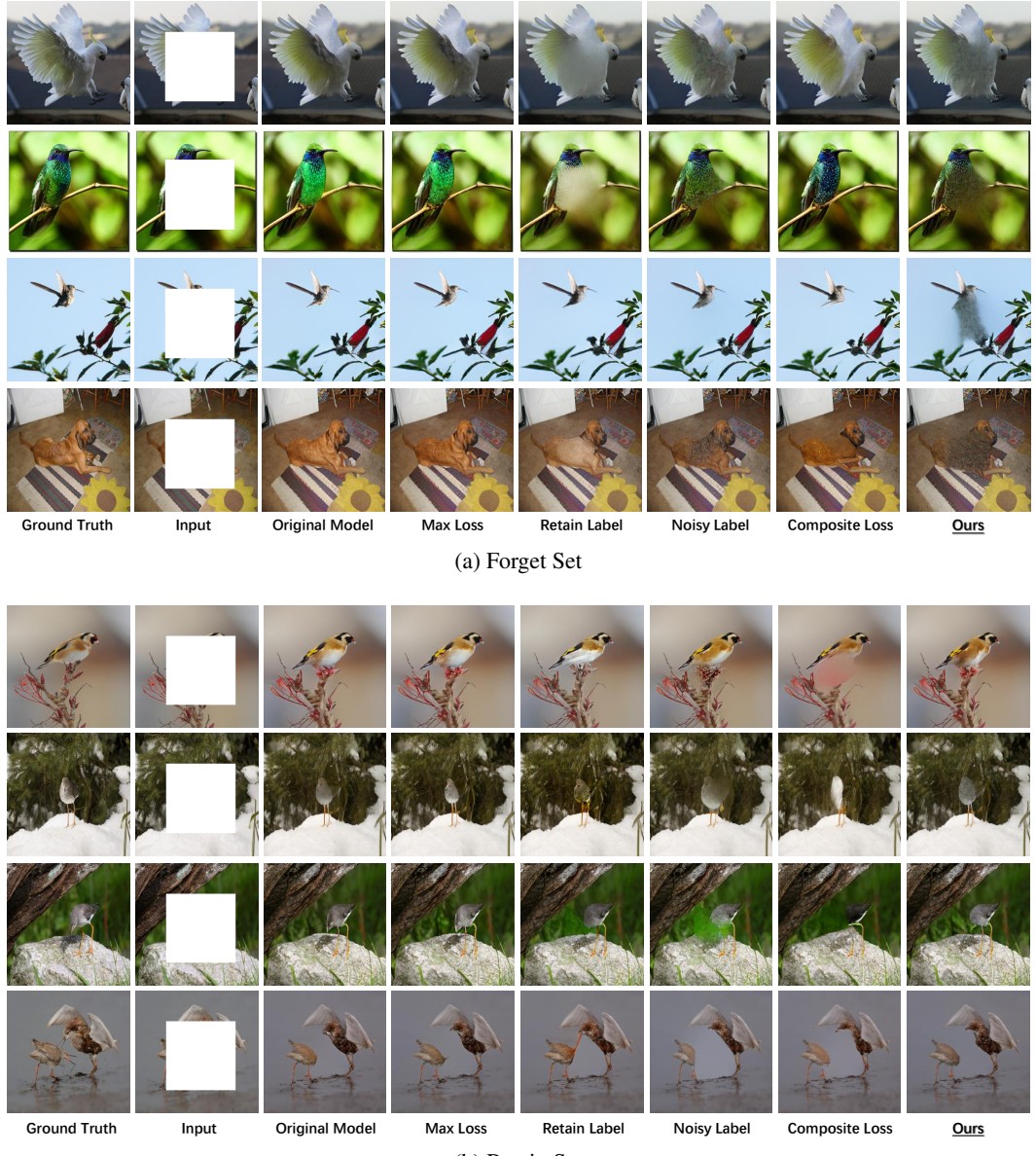

Figure 17: Generated images of cropping 25% at the center of the image. We crop the center 1/16 of the image. The upper half (a) is the forget set, and the lower half (b) is the retain set. For each set, we compare the performance of the baselines and our method on the inpainting task, where "Ours" represents the extreme case of the unlearning boundary in Phase I, that is, the point of highest degree of unlearning completeness.

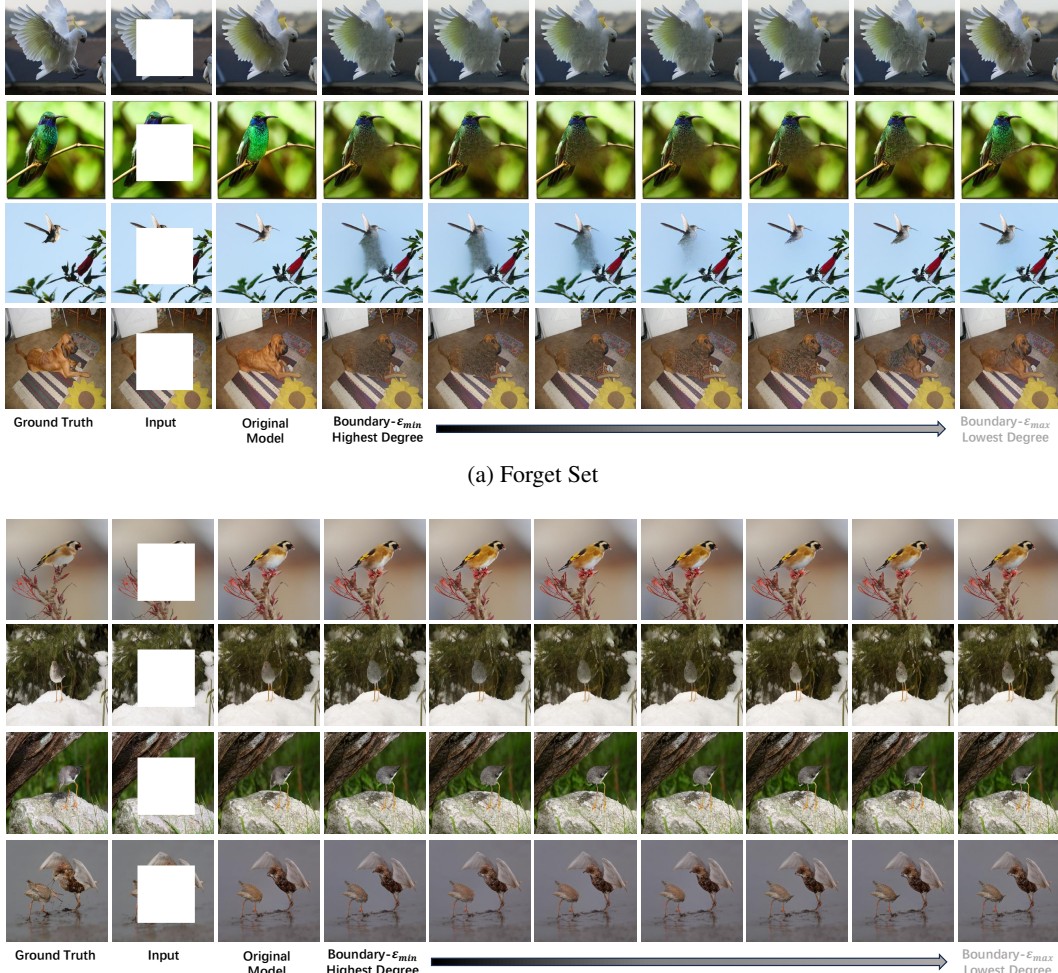

(a) Forget Set

(b) Retain Set

Figure 18: Generated images of cropping 50% at the center of the image under different degrees of unlearning completeness requirements. We crop the central 1/16 of the image. The upper half (a) represents the forget set, and the lower half (b) represents the retain set. For each section, we compare the effectiveness of our method's unlearning under different values of $\varepsilon$. Here, "Highest" and "Lowest" indicate the conditions of the highest and lowest degree of unlearning completeness, respectively.

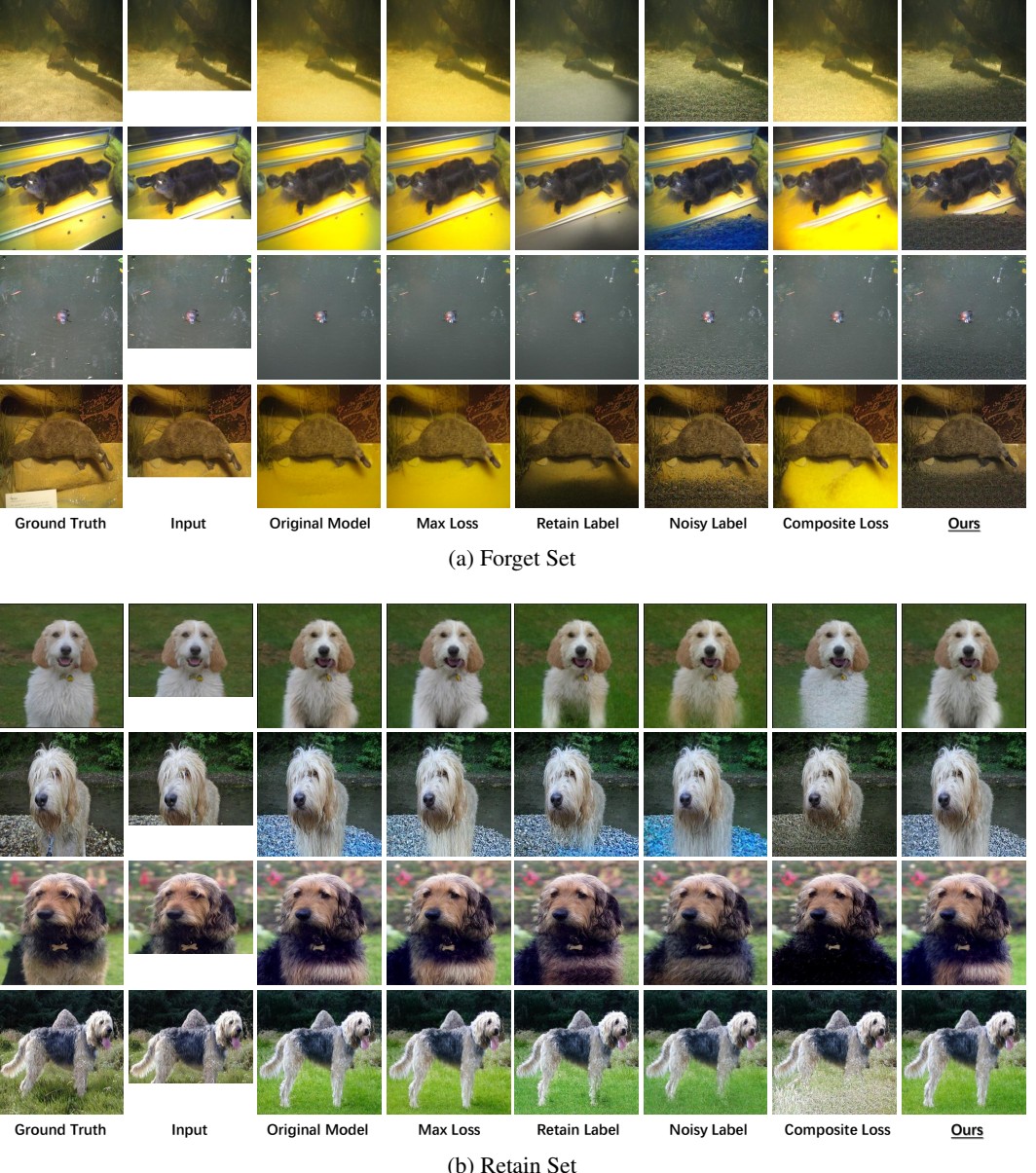

Figure 19: Downward extension by VQ-GAN. We crop the bottom 25% of the image. The upper half (a) is designated as the forget set, and the lower half (b) as the retain set. For each section, we compared the performance of the baselines and our method on the downward extension task, where "Ours" denotes the unlearning boundary condition in Phase I, that is, the point of highest degree of unlearning completeness.

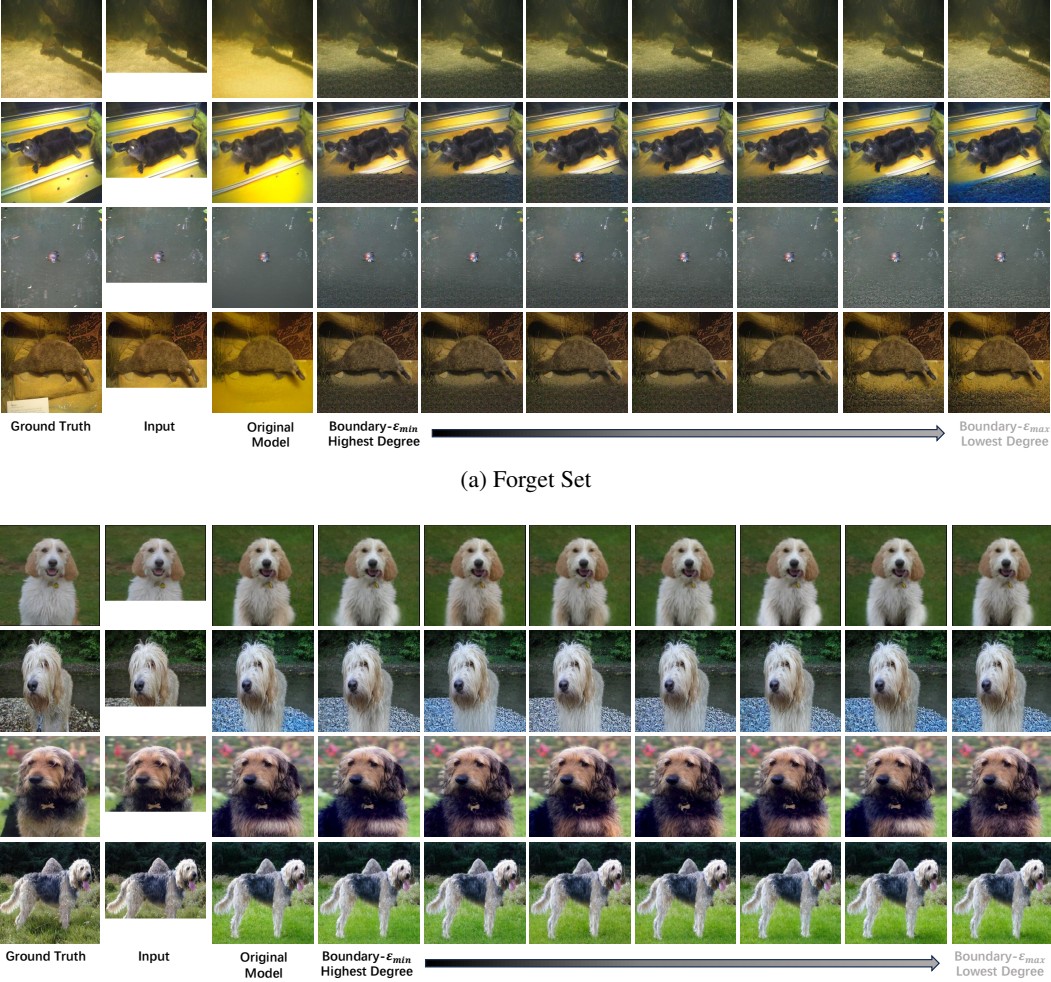

Figure 20: Downward extension by VQ-GAN under different degrees of unlearning completeness. We crop the bottom 25% of the image. The upper half (a) represents the forget set, and the lower half (b) represents the retain set. For each section, we compare the effectiveness of our method's unlearning under different values of $\varepsilon$. Here, "Highest" and "Lowest" indicate the conditions of the highest and lowest degree of unlearning completeness, respectively.

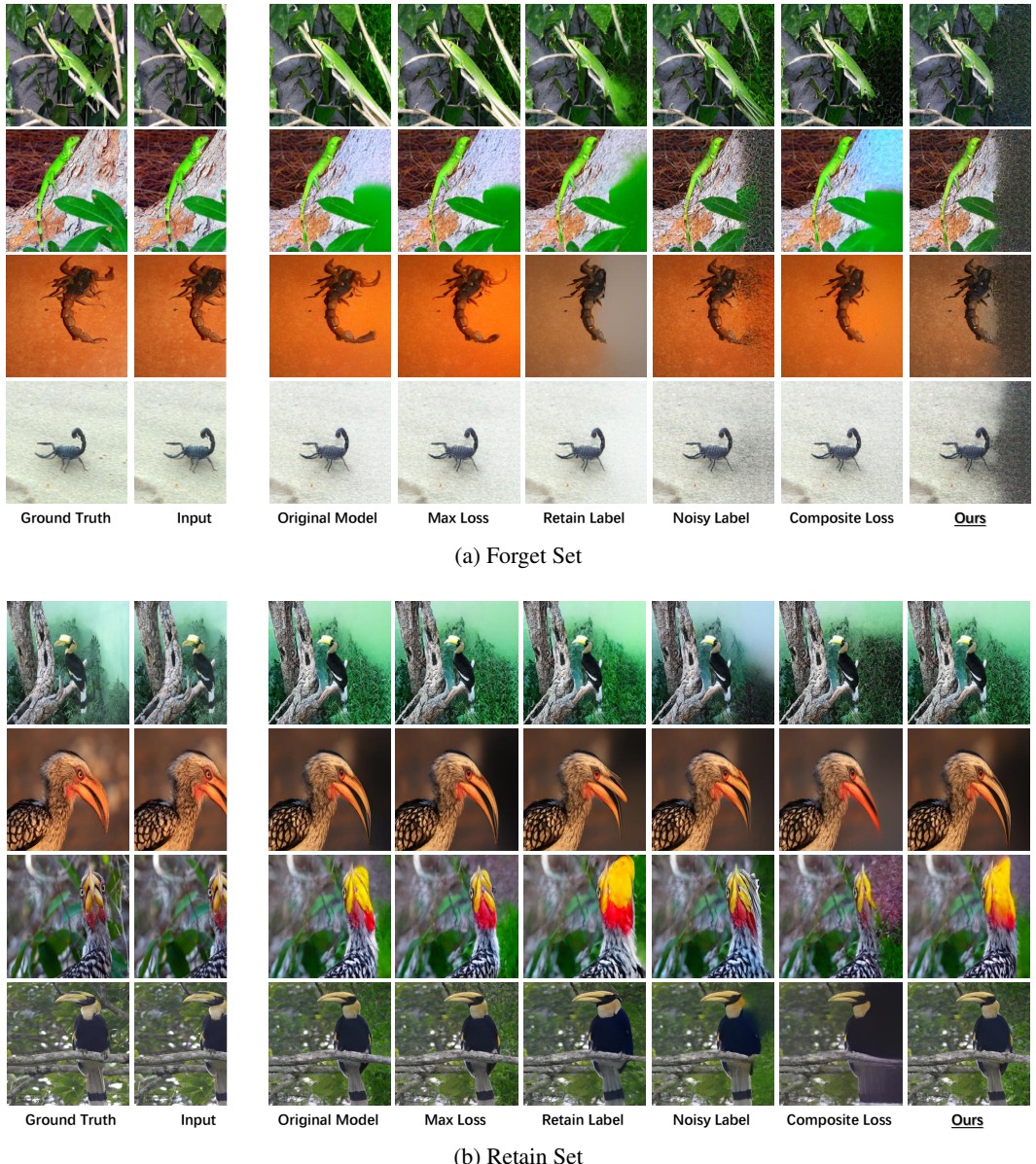

Figure 21: Rightward extension by VQ-GAN. We crop the right 25% of the image. The upper half (a) is designated as the forget set, and the lower half (b) as the retain set. For each section, we compared the performance of the baselines and our method on the rightward extension task, where "Ours" denotes the unlearning boundary condition in Phase I, that is, the point of highest degree of unlearning completeness.

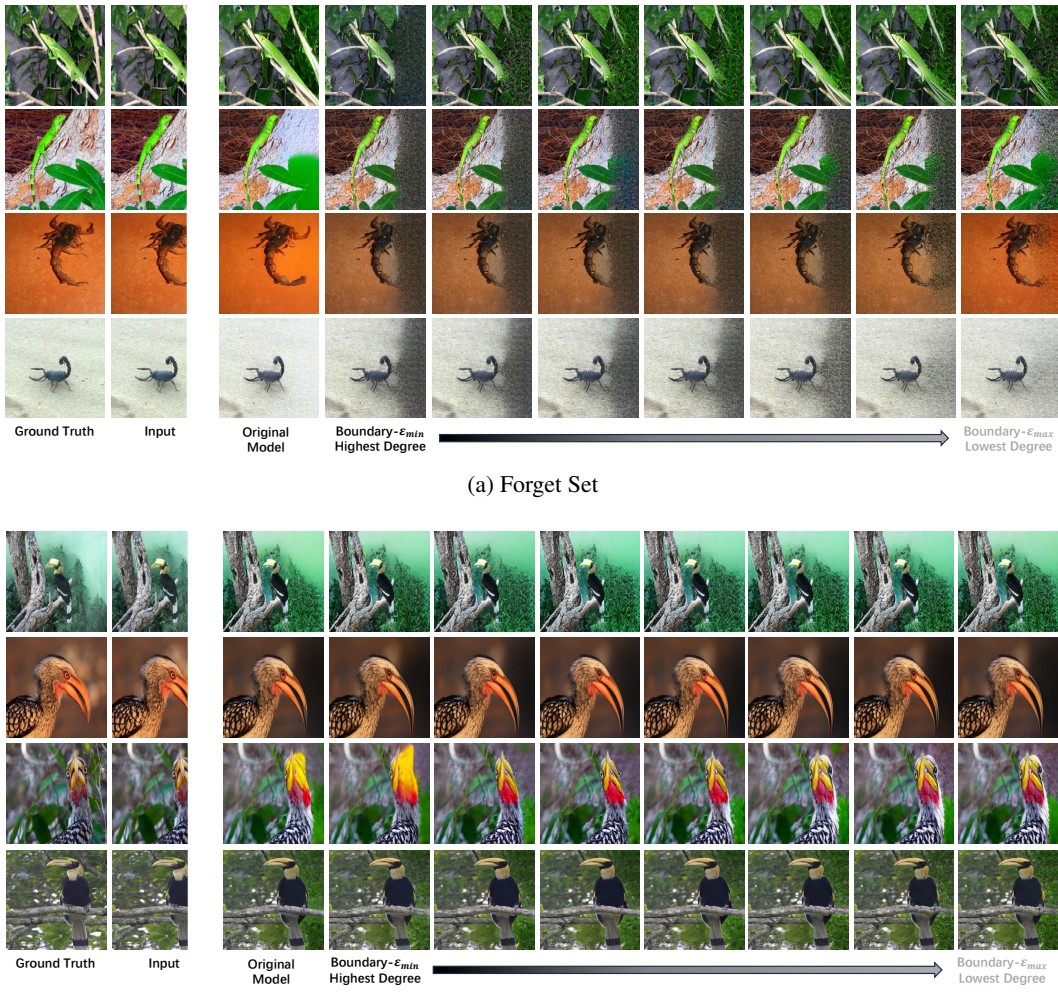

Figure 22: Rightward extension by VQ-GAN under different degrees of unlearning completeness. We crop the right 25% of the image. The upper half (a) represents the forget set, and the lower half (b) represents the retain set. For each section, we compare the effectiveness of our method's unlearning under different values of $\varepsilon$. Here, "Highest" and "Lowest" indicate the conditions of the highest and lowest degree of unlearning completeness, respectively.

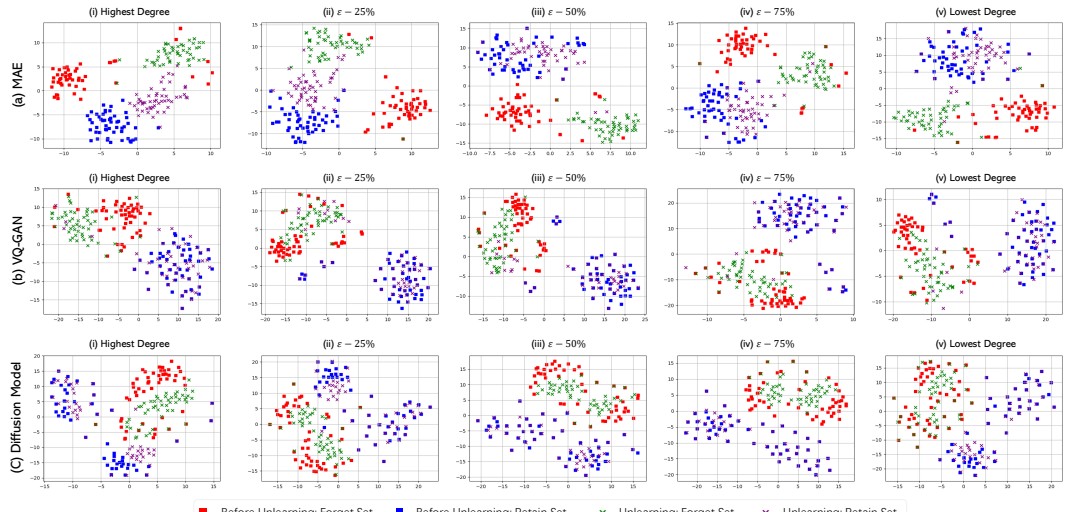

Figure 23: T-SNE analysis between images generated by our method and ground truth images under different degrees of unlearning completeness.

analysis on three mainstream I2I generative models. During the two different phases of controllable unlearning, we design the form of the control function $\psi(\theta)$ separately.

Specifically, in Phase I, we set $\psi(\theta) = \alpha\|\nabla f_1(\theta)\|^\delta$, where we test the convergence rates of $f_1(\theta)$ and $f_2(\theta)$, as well as the overall convergence rate, for $\delta = 1$, $\delta = 2$, $\delta = 3$, and $\delta = 4$. As shown in Figure 24, It is apparent that at Phase I for $c = 2$, that is $\psi(\theta) = \alpha\|\nabla f_1(\theta)\|^2$, the overall convergence rate is optimal.

In Phase II, we set $\psi(\theta) = \beta(f_1(\theta) - \varepsilon)^\delta$, where we tested the convergence rates for $\delta = 1$ and $\delta = 3$. Subsequently, we changed the form of $\psi(\theta)$ to $\psi(\theta) = \beta(f_1(\theta) - \varepsilon)^\delta\|\nabla f_1(\theta)\|^2$, and we tested the convergence rates for $\delta = 1$ and $\delta = 3$. Comparing the aforementioned scenarios, the overall optimal convergence rate in Phase II is obtained when $\psi(\theta) = \beta(f_1(\theta) - \varepsilon)^1\|\nabla f_1(\theta)\|^2$.

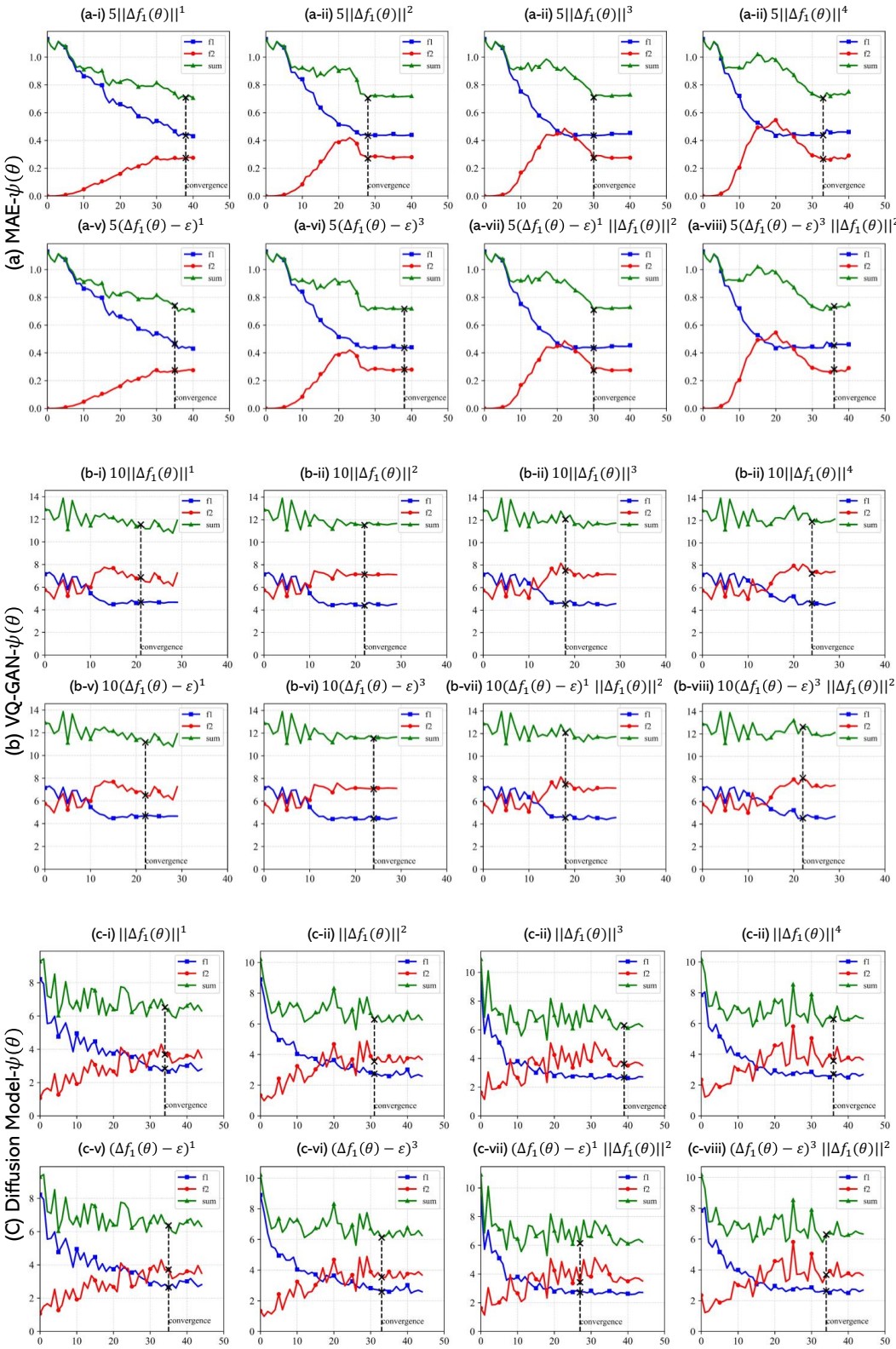

Figure 24: The convergence rates under different control functions $\psi(\theta)$. As illustrated in figure, include three sections: MAE, VQ-GAN, and the diffusion model. Each section contains two rows, corresponding to Phase I and Phase II, respectively. The titles on each subplot indicate the forms of the control function $\psi(\theta)$.

