# OpenReview forum: "Controllable Unlearning for Image-to-Image Generative Models via $\epsilon$-Constrained Optimization"
_ICLR.cc/2025/Conference — ICLR 2025 Poster_

### Official Review · Reviewer_hwTN · 2024-11-03

**Soundness:** 3
**Presentation:** 3
**Contribution:** 3
**Rating:** 6
**Confidence:** 3

**Summary:**

In this work, the authors study the problem of machine unlearning (MU) in image-to-image (I2I) generative models. Unlike prior studies, this approach diverges from a single objective to better consider the tradeoff between unlearning completeness and model utility, offering more flexibility for varying user needs. Specifically, the authors first reformulate the bi-objective MU problem into a constrained optimization problem and then propose a gradient-based algorithm to find Pareto optimal solutions. The proposed algorithm comes with a theoretical guarantee for convergence. Additionally, empirical results show that the proposed method provides a good balance between the two objectives, performing competitively among baselines.

**Strengths:**

1. The proposed algorithm is well-motivated and comes with a theoretical guarantee for convergence, laying a solid theoretical foundation for application.
2. The authors identify an overlooked issue in MU for I2I generative models in previous works: the failure to cater to varying user expectations in the real world, $i.e.,$ lack of controllability. Based on this observation, they derive a novel solution to this new bi-objective problem, which has practical significance for improving I2I generative models.
3. The empirical findings align with the theoretical results, demonstrating that the solutions found by the proposed algorithm achieve good performance in terms of both objectives, $i.e.,$ unlearning completeness and model utility.

**Weaknesses:**

1. The definition of unlearning completeness in the paper is problematic. The paper uses the KL divergence between distributions of forget data and reconstructed data to evaluate the completeness of unlearning, ultimately approximating it with the L2 loss. However, both losses are not ideal criteria for assessing unlearning performance, as they are defined in pixel space and disregard the original image manifold. Generative models can exploit this by outputting inconsistent pixel values when operating on the forget set, leading to suboptimal unlearning results. This is evident in the artifacts in reconstructed image examples from the forget set in Appendix F, $e.g.,$ in the inpainting task.
2. The proposed algorithm doubles memory usage, as it requires storing two separate model gradients. It also involves model-level gradient operations (as described in Algorithm 1, Line 8), making it more complex than other baselines. This can be less practical for larger models. A detailed computational complexity comparison and a discussion on memory usage would be helpful.
3. Confusing evaluation. In Table 1, the Inception Score (IS) appears in both columns for the forget set and retain set. It is unclear why IS should be "the less the better" (if that is the meaning of the down-arrow) for the forget set. If unlearning completeness is linked to low-quality generation, then the objective becomes trivial—a simple classifier to detect forget data would suffice. Echoing my previous point in W1, the generative model should at least produce a natural or similar image, even if the input is out-of-distribution. This requirement is completely overlooked here.
4. Minor typos. For example, in Equation 1, $I_\theta=D_\phi(E_\gamma(\mathcal{T}(x)))$ should be $I_\theta=D_\phi(E_\gamma(x))$ to be consistent with the rest of the text.

**Questions:**

Is the proposed method applicable to text-guided I2I generative models, such as image editing models?

---

> ### Author Response · Authors · 2024-11-14
>
> **1. Weaknesses:** The definition of unlearning completeness in the paper is problematic.
>
> **Response:** Thank you for your insights. This work focuses on improving [1]. Thus, we adhere to existing settings, conducting unlearning at the sample level. Our goal is to eliminate the influence of these samples on the model, or in other words, to erase the knowledge the model has learned from these samples. We fully agree with your point, particularly when assuming the unlearning target is a distribution, or an abstract concept. In such cases, the unlearning target should be the data manifold or a subspace within the data representation space, which could have broader applications in real-world scenarios, such as typical style erasure cases. This is an issue that requires further exploration in future research.
>
> Fortunately, our framework can be simply extended to accommodate this setting. For instance, as you mentioned, if the generative model's unlearning involves the original image manifold, we can shift the loss in the unlearning target from a pixel space metric to a manifold metric on the image, or introduce manifold regularization in the outputs, and then optimize and adjust based on our unlearning framework. Thank you again for your insightful suggestions, which are very helpful for improving our future work!
>
> **2. Weaknesses:** A detailed computational complexity comparison and a discussion on memory usage would be helpful.
>
> **Response:** Thank you for your suggestion. We conduct a brief analysis as follows. Assuming we use the Adam optimizer, in each iteration, the computational complexity for methods involving only a single model gradient (i.e., other baselines) arises from the following sources:
> 1. Loss computation: The forward pass to compute the loss has a complexity of $O(N×P)$, where $N$ is the number of samples and $P$ is the number of parameters.
> 2. Gradient computation: The backward pass to compute the gradient of the loss has a complexity of $O(N×P)$.
> 3. First moment update: Adam updates the first moment estimate (mean) $m_t$. This requires $O(P)$ operations, as each parameter's moment is updated using the computed gradient.
> 4. Second moment update: Similarly, updating the second moment estimate (variance) $v_t$ also requires $O(P)$ operations.
> 5. Parameter update: Adam uses the updated first and second moments to modify the parameters, which requires $O(P)$ operations.
>
> Thus, the total computational complexity for each iteration is: $O(N×P)$ (loss computation) + $O(N×P)$ (gradient computation) + $O(P)$ (moments and update) = $O(N×P)$.
>
> For each iteration in our algorithm, the computational requirements are:
> 1. Loss computation: The forward pass to compute both $f_1(\theta_t)$ and $f_2(\theta_t)$ requires two forward passes, each with a complexity of $O(N×P)$, so this step has a total complexity of $O(N×P)$.
> 2. Gradient computation: The backward pass to compute both $\nabla f_1(\theta_t)$ and $\nabla f_2(\theta_t)$ requires two backward passes, each with a complexity of $O(N×P)$, so this step also has a total complexity of $O(N×P)$.
> 3. First moment update: Adam updates the first moment estimate $m_t$ based on the combined gradient $g_t$, which requires $O(P)$ operations.
> 4. Second moment update: Adam also updates the second moment estimate $v_t$ based on the combined gradient $g_t$, requiring $O(P)$ operations.
> 5. Dual problem solution (Line 8): Solving the dual problem in Line 8 involves vector operations such as dot products and norms, which have a complexity of $O(P)$.
> 6. Parameter update: The model parameters are updated based on the Adam update rule, requiring $O(P)$ operations.
>
> In terms of computational complexity, although our method incurs slightly greater computational costs compared to other baselines, the overall computational complexity remains $O(N×P)$. Regarding memory usage, due to model-level gradient operations, we employ a strategy of trading time for space in the practical implementation to address this issue. Through our experiments, we have verified that our method can also be effectively applied to larger models, such as diffusion model, with computational efficiency that is entirely acceptable.
>
> **3. Weaknesses:** Confusing evaluation.
>
> **Response:** Your feedback is indeed meaningful. We adhere to the settings from prior work [1], which may place more emphasis on erasing sample-specific knowledge, potentially being too stringent in the context of distribution or concept unlearning. We concur with your point that future research on unlearning in generative models should be defined on data manifolds, as this aligns more closely with real-world needs.
>
> **4. Weaknesses:** Minor typos.
>
> **Response:** We apologize for the imperfect writing. We have carefully reviewed the paper and fixed identified typos.

---

> ### Author Response · Authors · 2024-11-14
>
> **5. Questions:** Is the proposed method applicable to text-guided I2I generative models, such as image editing models?
>
> **Response:** We deem this is feasible. In the current research on unlearning in image generation models, we categorize it into two types based on the unlearning target: one is unlearning a fixed set, viewed as sample-level unlearning, aimed at eliminating the influence of the forgotten set on the model. This scenario is common in I2I generation models and is the focus of this paper. The other is unlearning an abstract concept, typically seen in text-to-image diffusion models, where the goal is to prevent the model from generating images containing the concept after unlearning.
>
> For the latter, concept unlearning, existing research often defines concepts as corresponding text, such as "Van Gogh" representing the concept of Van Gogh's painting style, "Monet" for Monet's style, and "Elon Musk" for the entity concept of Musk. They achieve concept unlearning by defining concepts as text and ensuring the model cannot output images containing the concept given the textual condition. We believe that based on this definition, our method can be easily extended. For instance, preventing outputs that include the concept information under given conditions can be viewed as objective 1, while maintaining the quality of images under other conditions as objective 2. Our framework can precisely control the degree of concept unlearning.
>
> **Additional Discussion.** However, we contend that the existing definition may not fully align with our ideal unlearning target, reflected in two aspects:
>
> - First, defining concepts with text is not always suitable. Concepts are relative for humans, and textual definitions inherently carry ambiguity, as different textual expressions may correspond to the same concept. If it is necessary to define the concept of being forgotten using text, we believe it may require semantic alignment in the text encoder of the text-to-image generation model.
>
> - Second, much of the existing research on concept unlearning focuses on unlearning the mapping from text to image — that is, ensuring that the model cannot output images containing the forgotten concept given specific text. However, what we actually need is for the model to be unable to output images containing the forgotten concept, regardless of the text provided. In other words, we want the forgotten model to lack the capability to generate the concept under any given conditions.
>
> The core of these issues lies in the definition of concepts. We argue that concepts should be defined as a data manifold, a distribution, or a subspace, as pointed out by Wang et al. in [2]. Concept unlearning should first identify the concept to be forgotten and then excise it from the original knowledge. For example, if a concept is defined as a distribution $p_1$, and the original model learns the data distribution $p_0$, then concept erasure in a text-to-image model should first determine distribution $p_1$, then determine the distribution $p_0'$ after removing $p_1$ from $p_0$, and finally solve a distributional shift problem from $p_0$ to $p_0'$ to achieve concept unlearning. Under this definition, concept unlearning in text-to-image models might not be as straightforward for our method to extend. However, based on our understanding, flow-based models like the Wasserstein gradient flow could be a potential approach to address this issue, marking a possible future exploration direction.
>
> Considering more complex scenarios, as pointed out in [3], if there are connections between concepts such as a direct pathway in the model’s weight units that can activate the forgotten concept, or activation through a combination of several other concepts, then merely unlearning the direct pathway to this concept may not be sufficient. We also need to remove the connections from other concepts to the forgotten concept. We believe this represents a more complex class of problems, involving the interpretability of the generative models.
>
> [1] Li, Guihong, et al. "Machine unlearning for image-to-image generative models." arXiv preprint arXiv:2402.00351 (2024).
>
> [2] Wang, Peng, et al. "Diffusion models learn low-dimensional distributions via subspace clustering." arXiv preprint arXiv:2409.02426 (2024).
>
> [3] Shumailov, Ilia, et al. "Ununlearning: Unlearning is not sufficient for content regulation in advanced generative ai." arXiv preprint arXiv:2407.00106 (2024).

---

### Official Review · Reviewer_icG6 · 2024-11-04

**Soundness:** 3
**Presentation:** 3
**Contribution:** 3
**Rating:** 6
**Confidence:** 3

**Summary:**

This submission formulates the controllable I2I unlearning problem as a $\epsilon$-constrained problem, which differs from the prior objective. By reformulating the problem as a $\epsilon$-constrained bi-objective function, two Pareto optimal solutions and the valid range of the control coefficient $\epsilon$ can be obtained. Furthermore, the authors provide a theoretical analysis of the convergence of the proposed method under various control functions used to govern the direction of parameter updates. The experimental results on two well-known benchmarks show the effectiveness over the mentioned baselines.

**Strengths:**

- The proposed method is sound. The proposed method reformulates the I2I unlearning problem by integrating the $\epsilon$-constrained method which is widely used in multi-objective optimization. This integration makes the unlearning degree controllable and brings a few theoretical merits, such as convergence analysis.
- This submission is well written and organized, which reduces the difficulty in reading and comprehending.

**Weaknesses:**

- This could be an improvement of [1] based on $\epsilon$-constrained method. Technically, please provide the specific design of $\epsilon$-constrained optimization for the I2I unlearning problem. And why $\epsilon$-constrained method is required to integrate with the I2I unlearning problem?
- Some claims are not evaluated. For instance, in line 70,  how the challenge ``First and foremost, this approach offers a solitary resolution,..’’ is addressed?
- Evaluation of different crop sizes should be conducted. In practice, not only the degree of forgetting but also the size of crop area is defined by users.

[1] Machine unlearning for image-to-image generative models, ICLR 2024

**Questions:**

- Why the results of Composite Loss is different from those reported in [1]? Please provide more details of implementation differences about it.
- According to Fig.4, why the visualization of the retained set of MAE is changed after unlearning? This is quite different from [1].
- Can you provide experimental results to demonstrate the proposed enjoys better unlearning efficacy than other methods? The theoretical results sometimes are different from real practice.

[1] Machine unlearning for image-to-image generative models, ICLR 2024

---

> ### Author Response · Authors · 2024-11-14
>
> **1. Weaknesses:** Why the $\varepsilon$-constrained method is required to integrate with the I2I unlearning problem? Please provide the specific design of $\varepsilon$-constrained optimization for the I2I unlearning problem.
>
> **Response:** We apologize for the lack of clarity in our previous statements. We first explain why applying $\varepsilon$-constrained optimization to I2I unlearning, primarily based on the following three reasons:
>
> - Initially, the original unlearning problem is defined as a bi-objective optimization problem (i.e., Eq.3), where the unlearning objective and the objective of preserving model performance are considered equally important. However, this is not the case in practical applications. In real-world scenarios, we often prioritize the unlearning objective, or the unlearning objective is often a hard constraint, such as regulations imposed by governments. Therefore, we aim to satisfy the unlearning constraints first, before improving model performance.
>
> - In the real world, the requirements for unlearning standards vary among different individuals or institutions. We aim to develop a method that allows precise control over the degree of unlearning.
>
> - Existing methods often require tedious parameter tuning during implementation. We aim to avoid this. In our approach, simply adjusting the form of optimization allows us to determine the boundaries for the hyperparameter $\varepsilon$, thereby clarifying the effective range of hyperparameter values.
>
> Based on the reasons above, introducing $\varepsilon$-constrained optimization into I2I unlearning effectively addresses these objectives. Furthermore, theoretically, $\varepsilon$-constrained optimization is equivalent to the original bi-objective optimization problem.
>
> **2. Weaknesses:** Some claims are not evaluated, such as line 70.
>
> **Response:** We sincerely apologize for any confusion caused by our inadequate expression. In line 70, we initially used "solitary resolution" to convey the meaning of "fixed result." To avoid misunderstanding, we have replaced it with "fixed result".
>
> **3. Weaknesses:** Evaluation of different crop sizes should be conducted.
>
> **Response:** Due to space constraints, we report this experiment in Appendix G.1 and G.3.
>
> **4. Questions:** Why the results of Composite Loss is different from those reported in [1]?
>
> **Response:** We replicate their experiments on our server, utilizing the hyperparameters recommended in the original paper [1], such as learning rate and optimizer parameters. To facilitate a fair comparison, for other experimental details, we adopt the same setup for all compared methods.
>
> As for the differences from the results in [1], we deem they are due to certain experimental settings that we could not fully align with theirs. Specifically, due to limitations in experimental conditions, some parameters like batch size, epochs, and multi-GPU training setups may not exactly match theirs.
> Additionally, certain experimental configurations, such as the random seed, could not be aligned with [1] as they have not been disclosed. Furthermore, we do not apply any data augmentation techniques during the experiments, and correspondingly, our method also does not utilize such operations either. For more details, we have reported the experimental settings in Appendix C.
>
> **5. Questions:** Why the visualization of the retained set of MAE is changed after unlearning? This is quite different from [1].
>
> **Response:** The issue you pointed out does indeed exist. When performing the unlearning operation, [1] updates the encoder's parameters through the L2 loss between encoders, whereas our method achieves updates via the L2 loss of the outputs (i.e., freezing the decoder and updating the encoder). This pixel-level loss results in greater fluctuations in the loss for models with weaker generative capabilities, impacting the encoder. Compared to VQ-GAN and Diffusion, MAE has significantly weaker generative capabilities, which is why this difference arises.
>
> **6. Questions:** Can you provide experimental results to demonstrate the proposed enjoys better unlearning efficacy than other methods?
>
> **Response:** In Table 1 of the main text, we compare the performance of our method with other methods under the highest degree of unlearning completeness, ensuring that the hyperparameters for the other methods are consistent with those in [1]. Due to space constraints, we have reported the remaining experimental results in Appendix E. Extensive experimental results validate the effectiveness of our method, demonstrating that we surpass existing baselines in terms of both unlearning efficacy and the maintenance of model performance.
>
>
> [1] Li, Guihong, et al. "Machine unlearning for image-to-image generative models." arXiv preprint arXiv:2402.00351 (2024).

---

> ### Comment · Reviewer_icG6 · 2024-11-22
>
> Thank you. Most of my concerns have been addressed. I still have a question about the problem modeling in this submission. The description in Sec. 4.2 suggests that the bi-level unlearning optimization problem could be formulated with the $\epsilon$-constraint methods in multiobjective optimization. To solve such a problem, the authors adopt a special variant of Sequential Quadratic Programming (SQP)[1]. I have not found any special design for formulating the unlearning optimization problem. Can I consider the controllable unlearning framework to be the key technical contribution of this submission?
>
> [1] Numerical optimization: theoretical and practical aspects, Springer Science & Business Media, 2006.

---

> ### Comment · Reviewer_icG6 · 2024-11-22
>
> Thank you! I have acknowledged your motivation to reformulate the unlearning optimization problem with $\epsilon$-constraint method. From my perspective, the ''two more technical contributions'' should be the properties of the proposed reformulation. In general, I am happy to maintain my positive score.

---

### Official Review · Reviewer_NJMm · 2024-11-04

**Soundness:** 3
**Presentation:** 4
**Contribution:** 3
**Rating:** 8
**Confidence:** 4

**Summary:**

The paper proposes a machine unlearning approach for generative image-to-image models. Machine unlearning algorithm in generative domain aims to make the model forget a specific subset samples (for e.g. defined by classes) while retaining its generalization capability on the other samples in order to address issues related to privacy and biases. The paper proposes a controllable unlearning algorithm flexible enough to balance between quality/degree of unlearning concepts and the model’s generalization capabilities. The approach uses a gradient based method to solve a constraint optimization objective where the constrain is to forget a certain specified set while retaining its reconstruction quality on remaining samples. The paper also shows theoretical analysis of its approach using Pareto optimality. The paper shows quantitative and qualitative results on in-painting/out-painting tasks to demonstrate the efficacy of the proposed approach.

**Strengths:**

The paper explores an unlearning approach for generative image-to-image models that uses gradient based method to solve a constrained optimization objective. The paper explains the issues present in the current machine unlearning domain and address these issues using a controllable optimization where the users have control over the unlearning optimization (model unlearning while maintaining model generalization). The proposed framework shows better results on ImageNet-1k and Places-365 dataset for in-painting tasks compared to other baseline unlearning approaches. The paper provides detailed ablation experiments and theoretical analysis to explain its proposed algorithm. The paper is well-written, easy to follow and contains a pseudocode that explains the methodology clearly.

**Weaknesses:**

It would helpful for the reader to see some discussions around the robustness of the concepts removal. For example is it possible to use some attack that resurfaces the forget set, for example as shown in paper Petsiuk, Vitali, and Kate Saenko. "Concept Arithmetics for Circumventing Concept Inhibition in Diffusion Models." arXiv preprint arXiv:2404.13706 (2024).
It would be helpful for the readers if some more related unlearning papers are added as references:

[1] Petsiuk, Vitali, and Kate Saenko. "Concept Arithmetics for Circumventing Concept Inhibition in Diffusion Models." arXiv preprint arXiv:2404.13706 (2024)

[2] Kumari, Nupur, et al. "Ablating concepts in text-to-image diffusion models." Proceedings of the IEEE/CVF International Conference on Computer Vision. 2023.

**Questions:**

It would be helpful if the paper can answer/comment on the following question/suggestion:

1. Is it possible to formulate the unlearning objective that simply in-paints with background content ( i.e. instead of predicting a gaussian type patch in the image for in-painting task, the model predicts the background and does not generate the subject that is to be forgotten). Does this require modification in the formulation that uses Divergence(P_Xf | N(0, sigma)) as condition.

---

> ### Author Response · Authors · 2024-11-14
>
> **1. Weaknesses:** Some discussions around the robustness of the concept removal would be helpful for readers.
>
> **Response:** Thank you for your suggestions. We agree that discussing the robustness of concept removal is indeed essential. For concept unlearning within text-to-image models, where the target for unlearning is an abstract concept, such as the "Van Gogh" style or an entity concept like "Musk," the goal is to ensure that the model's output images do not contain the specified concept. In such cases, the robustness of unlearning can be validated by determining whether it is possible to restore the concept (e.g., through attacking).
>
> However, this paper focuses on unlearning a set of samples or a distribution, meaning eliminating the knowledge the model has learned from these samples so that it cannot reconstruct this knowledge under any input conditions. In this scenario, we cannot directly verify through attackers as concept unlearning. Instead, we follow the setup of [1], simulating input variability by altering aspects of the input image, such as adjusting the crop region’s position or scale, to validate the robustness of unlearning. The results have been reported in Appendix G.1. Additionally, we believe your insights significantly contribute to our research, and incorporating a discussion on this issue will aid readers in better understanding. Therefore, we have added a corresponding discussion in Appendix G.1 and supplemented Appendix G.1 with all the references you mentioned (i.e., [2][3]).
>
> **Additional Discussion.** Regarding your point on concept unlearning, we agree that discussing its robustness is crucial and is a key area for future research. Current methods for concept unlearning have not yet achieved the desired robustness due to three main reasons:
>
> - First, these methods typically define concepts using text in text-to-image generation models. However, concepts are inherently relative for humans, and defining them textually introduces ambiguity since different textual expressions may represent the same concept to people.
>
> - Second, most existing work on concept unlearning focuses on breaking the mapping from text to image, which means that given certain text, the model fails to generate images containing the forgotten concept. However, what is actually needed is that the model should be unable to generate images containing the forgotten concept under any text input.
>
> - Third, considering the interconnections between concepts, where one concept might be composed of several other concepts, simply unlearning the specified concept may not be sufficient.
>
> Therefore, we believe that focusing on the robustness of concept unlearning is essential.
>
> **2. Questions:** Can inpainting an image (using background content) be used as a substitute for the unlearning target?
>
> **Response:** We deem that describing the unlearning target as inpainting an image using only background content is feasible to some extent, such as concept unlearning. For instance, if we aim to protect privacy by unlearning parts of an image generation model that contain personal information (i.e., an abstract concept), we can first identify the region of the image containing such information, then simply mask this region, and subsequently generate a new image through inpainting, ensuring that the model’s output aligns with the inpainted new image. However, this approach has two issues:
>
> - Firstly, it must be ensured that the new image generated through inpainting does not contain the information that needs to be forgotten. We believe this can be accomplished by incorporating an additional adversarial discriminator using GAN training strategies or by employing reinforcement strategies.
>
> - Secondly, aligning the model's output with the inpainted new image merely confuses the knowledge learned by the model, increasing uncertainty during generation, which constitutes a superficial form of unlearning. However, based on our experimental experience, if the goal is merely to erase the influence of certain samples on the model, directly aligning with Gaussian noise may yield a more pronounced unlearning effect.
>
> [1] Li, Guihong, et al. "Machine unlearning for image-to-image generative models." arXiv preprint arXiv:2402.00351 (2024).
>
> [2] Petsiuk, Vitali, and Kate Saenko. "Concept Arithmetics for Circumventing Concept Inhibition in Diffusion Models." arXiv preprint arXiv:2404.13706 (2024).
>
> [3] Kumari, Nupur, et al. "Ablating concepts in text-to-image diffusion models." Proceedings of the IEEE/CVF International Conference on Computer Vision. 2023.

---

> > ### Comment · Reviewer_NJMm · 2024-11-23
> > **Comment response**
> >
> > Thank you for your response on the robustness and in-painting objective, it would be great to see this being referred in the manuscript too. I would like to maintain my rating.

---

> > > ### Author Response · Authors · 2024-11-24
> > >
> > > Thank you once again for your valuable comments. We have added the response to Weaknesses in Section 2.2 and Appendix H.1, and the response to Questions is included in Appendix B.

---

### Author Response · Authors · 2024-11-14

We sincerely thank all the reviewers for their valuable comments and suggestions, which are crucial for improving our work. We hope our response addresses your concerns.

---

### Meta-Review · Area_Chair_x9GP · 2024-12-21

**Metareview:**

This paper proposes a novel \epsilon constrained optimization problem to trade off unlearning and model utility in model unlearning, which is different from previous methods that consider only a single objective. The proposed method is supported by both theoretical guarantees and experimental results on two benchmark datasets. Given the positive feedback from all reviewers, I recommend accepting this paper.

**Additional Comments On Reviewer Discussion:**

All reviewers are generally positive about this paper. The discussion is around some technical details, and the authors provided detailed feedback in the rebuttal. The authors provided more discussions on the difference to previous methods and added more experiments in the appendix. Also, the computational complexity of the proposed algorithm is explained in detail. I weighted 70% on the ratings provided by the reviewers and 30% for the improved points in the rebuttal.

---

### Decision · Program_Chairs · 2025-01-22

Accept (Poster)